

# Estimation of fossil-fuel $CO_2$ emissions using satellite measurements of "proxy" species

Igor B. Konovalov[1], Evgeny V. Berezin[1], Philippe Ciais[2], Grégoire Broquet[2], Ruslan B. Zhuravlev[1], Greet Janssens-Maenhout[3]

[1]Institute of Applied Physics, Russian Academy of Sciences, Nizhniy Novgorod, Russia
[2]Laboratoire des Sciences du Climat et l'Environnement (LSCE/IPSL), CNRS-CEA-UVSQ, Centre d'Etudes Orme des Merisiers, Gif sur Yvette, France
[3]Joint Research Center, Institute for Environment and Sustainability, Ispra (Va), Italy

*Correspondence to*: Igor B. Konovalov (konov@appl.sci-nnov.ru)

**Abstract.** Fossil fuel (FF) burning releases carbon dioxide ($CO_2$) together with many other chemical species, some of which, such as, e.g., nitrogen dioxide ($NO_2$) and carbon monoxide (CO), are routinely monitored from space. This study examines the feasibility of estimation of FF $CO_2$ emissions from large industrial regions by using $NO_2$ and CO column retrievals from satellite measurements in combination with simulations by a mesoscale chemistry transport model (CTM). To this end, an inverse modeling method is developed that allows estimating FF $CO_2$ emissions from different sectors of the economy, as well as the total $CO_2$ emissions, in a given region. The key steps of the method are (1) inferring "top-down" estimates of the regional budget of anthropogenic emissions of proxy species (that is, $NO_x$ and CO in the case considered) from satellite measurements without using formal a priori constraints on these budgets, (2) application of emission factors (the NOx-to-$CO_2$ and CO-to-$CO_2$ emission ratios in each sector) that relate FF $CO_2$ emissions to the emissions of the proxy species and are evaluated by using data of "bottom-up" emission inventories, (3) cross-validation and optimal combination of the estimates of $CO_2$ emission budgets derived from measurements of the different proxy species. Uncertainties in the top-down estimates of the emissions of the proxy species are evaluated and systematic differences between the measured and simulated data are taken into account by using original robust techniques validated with synthetic data. To examine the potential of the method, it was applied to the budget of emissions for a western European region including 12 countries by using $NO_2$ and CO column amounts retrieved from, respectively, the OMI and IASI satellite measurements and simulated by the CHIMERE mesoscale CTM, along with the emission conversion factors based on the EDGAR v4.2 emission inventory. The analysis was focused on evaluation of the uncertainty levels for the "top-down" $NO_x$ and CO emission estimates and "hybrid" estimates (that is, those based on both atmospheric measurements of a given proxy species and respective "bottom-up" emission inventory data) of FF $CO_2$ emissions, as well as on examining consistency between the FF $CO_2$ emission estimates derived from measurements of the different proxy species. It is found that $NO_2$ measurements can provide much stronger constraints to the total annual FF $CO_2$ emissions in the study region than CO measurements, the accuracy of the $NO_2$-measurement-based $CO_2$ emission estimate being mostly limited by the uncertainty in the top-down $NO_x$ emission estimate. Nonetheless, CO measurements are also found to be useful as they provide additional constraints to $CO_2$ emissions and





enable evaluation of the hybrid FF $CO_2$ emission estimates obtained from $NO_2$ measurements. Our most reliable estimate for the total annual FF $CO_2$ emissions in the study region in 2008 (2.71±0.30 Pg $CO_2$) is found to be about 11 % and 5 % lower than the respective estimates based on the EDGAR v.4.2 (3.03 Pg $CO_2$) and CDIAC (2.86 Pg $CO_2$) emission inventories, with the difference between our estimate and the CDIAC inventory data being not statistically significant. In general, the

results of this study indicate that the proposed method has a potential to become a useful tool for identification of possible biases and/or inconsistencies in the bottom-up emission inventory data regarding $CO_2$, $NO_x$ and CO emissions from fossil fuel burning in different regions of the world.

## 1 Introduction

Carbon dioxide ($CO_2$) is commonly recognized as the major greenhouse gas whose increase is the driving force of recent and

future climate change (IPCC, 2013). Its atmospheric concentration has considerably increased (by 40 %) since the industrial revolution (Petit et al., 1999). This increase, the rate of which has accelerated in a recent decade, is attributed mostly to anthropogenic sources, such as fossil fuel (FF) burning (Canadell et al., 2007). Curbing further growth of $CO_2$ concentration has become a goal of international agreements such as the Kyoto protocol (UNFCCC, 1998) and the Paris Agreement on Climate Change (UNFCCC, 2015). In the latter, almost all countries are on board in an equal setting, and so 95% of the 2014

global $CO_2$ emissions are the starting point. Thus accurate knowledge of anthropogenic $CO_2$ emissions is of paramount importance both for climate prediction and mitigation policy purposes.

Over the past few decades a lot of efforts have been put into compilation of global (e.g., Olivier et al., 2005; GCP, 2010; Ciais et al., 2010a; Janssens-Maenhout et al., 2015) as well as regional (Gurney et al., 2009; Huang et al., 2011; Kurokawa et al., 2013; Zhao et al., 2012; Wang et al., 2013) inventories of $CO_2$ emissions from FF burning and other smaller

anthropogenic sources (such as, e.g., biofuel burning and cement production). Those emission inventories are based on available statistical information regarding economic activities and corresponding technologies. However, it is known that such information can be subject to errors and biases leading to considerable uncertainties in emission estimates, especially in the case of rapidly growing developing economies (e.g., Akimoto et al., 2006; Guan et al., 2012, Korsbakken et al., 2016). For example, the uncertainty of available estimates of the total FF $CO_2$ emissions in China is assessed to be about 15-20%

(Gregg et al., 2008). Much larger uncertainties may be associated with the sub-national spatial distributions and temporal evolution of FF $CO_2$ emissions within a year (Ciais et al., 2010b). The uncertainties in anthropogenic $CO_2$ emission inventory data are mostly due to inaccuracies of available data regarding fuel consumption and fuel chemical composition. Note that estimation of uncertainty in emission inventory data is itself a challenging task: in particular, as different inventories are usually based (at least partly) on common sources of information, their inter-comparison does not necessarily

result in revealing all the uncertainties.

A promising alternative approach to constrain $CO_2$ emissions and to assess the uncertainty in available emission estimates is inverse modeling (Enting, 2002); the key idea of this approach is to derive emission estimates from atmospheric





measurement data by optimizing emission parameters of a transport model. Such estimates are frequently referred to as "top-down", by contrast with "bottom-up" ones based on emission inventories alone. Numerous studies have successfully used in situ $CO_2$ measurements in the framework of this approach to constrain surface $CO_2$ fluxes associated mostly with biospheric and oceanic sources and sinks of $CO_2$ in different regions of the world (e.g., Gurney et al., 2002, Baker et al., 2006, Schulze

et al., 2009, Chevallier et al., 2010; Broquet et al., 2013). More recently, it was demonstrated that uncertainties in $CO_2$ flux estimates can be potentially reduced by using satellite $CO_2$ measurements (e.g., Chevallier et al., 2007; Houweling et al., 2004; Hungershoefer et al., 2010; Kadygrov et al., 2009; Nassar et al., 2011; Maksyutov et al. 2013; Reuter et al. 2014a). However, less progress has been made in isolating FF $CO_2$ emissions from other sources and sinks. Major limitations are due to the facts that the atmospheric variability of $CO_2$ is strongly affected by biogenic sources and sinks, such as plant

respiration and photosynthesis, and that the signatures of regional FF $CO_2$ emissions in $CO_2$ observations are typically weak relative to regional background $CO_2$ concentration, except near hot spots. Promising approaches suggest separation of FF $CO_2$ emissions from biospheric fluxes by using available measurements of radiocarbon content (14C) of $CO_2$ (e.g., Turnbull et al., 2009; Miller et al., 2012; Lehman et al., 2013; Basu et al., 2016), ground based $CO_2$ measurements in vicinity of strong anthropogenic emission sources like megacities (Bréon et al., 2015), or satellite $CO_2$ retrievals with sampling near

hot-spots (Bovensmann et al., 2010; Silva et al., 2013; Reuter et al., 2014b). However, neither of these approaches has already been sufficiently generalized to provide reliable estimates of the budget of anthropogenic $CO_2$ emissions in an arbitrary industrialized region of the world.

It has also been suggested that anthropogenic $CO_2$ emissions can be constrained to a certain extent by measurements of "proxy" species, whose sources are mostly collocated in time and space with $CO_2$ sources (Rivier et al., 2006;

Suntharalingam, 2004). The measurements of proxy species can be either combined with $CO_2$ measurements (Palmer et al., 2006; Rivier et al., 2006; Suntharalingam, 2004; Brioude et al., 2012) or used alone but with information on a relationship between emissions of $CO_2$ and of the proxy species from bottom-up emission inventories. In the second approach, Berezin et al. (2013) estimated multiannual relative changes of FF $CO_2$ emissions from China by using satellite measurements of nitrogen dioxide ($NO_2$) and emission inventory data on the ratio of FF emissions of $CO_2$ and nitrogen oxides ($NO_x$). A

similar approach was employed by Konovalov et al. (2014) to obtain estimates of $CO_2$ emissions from biomass burning in Siberia by using satellite measurements of carbon monoxide (CO) and of aerosol optical depth.

The goal of this study is to examine the feasibility of inferring estimates of annual budgets of $CO_2$ emissions from fossil fuel burning in a given industrialized region with a typical size of the order of 1000 km by using satellite measurements of $NO_2$ and CO. In doing so, we develop a special method by building upon the ideas that were exploited in Berezin et al. (2013) and

Konovalov et al. (2014). The method includes several major steps, namely: (1) inferring "top-down" estimates of total anthropogenic emissions of the proxy species from satellite measurements by using simulations performed with a mesoscale chemistry transport model (CTM), (2) applying NOx-to-$CO_2$ (or CO-to-$CO_2$) emission conversion factors given by "bottom-up" emission inventories to relate FF $CO_2$ emissions to the anthropogenic emissions of the proxy species from the previous





step, (3) cross-validation and optimal combination of estimates of the FF $CO_2$ emission budgets derived from measurements of different proxy species. As a result, we obtain a "hybrid" FF $CO_2$ emission estimate integrating information coming from measurements and bottom up inventories. The use of $NO_x$ and CO as proxy species in the context of our approach is justified because their satellite measurements are known to contain a strong signal associated with human activities in industrial regions and have abundantly been used earlier to constrain emissions of, respectively, nitrogen oxides (e.g., Martin et al., 2003; Konovalov et al., 2006; Napelenok et al., 2008; Miyazaki et al., 2012; Gu et al., 2014) and CO (e.g., Arellano, 2004; Pétron, 2004; Kopacz et al., 2010; Hooghiemstra et al., 2012; Krol et al., 2013; Jiang et al., 2015) from various sources, including anthropogenic ones. Note that although $NO_x$ and CO emissions from FF burning are more sensitive to technological factors than $CO_2$ emissions, different aspects of the combustion technology are expected to affect $NO_x$ and CO emissions in different ways: e.g., while $NO_x$ emissions are strongly dependent on the temperature of combustion (more $NO_x$ is released at higher temperatures), CO emissions can be regarded as a measure of the incompleteness of combustion processes. So, the combination of hybrid FF $CO_2$ emission estimates derived from both $NO_2$ and CO measurements can enable a compensation of a part of the uncertainties associated with inaccurate knowledge of technology and conditions of combustion affecting separately $NO_2$- and CO measurement based FF $CO_2$ emission estimates.

Particular efforts in this study were made to provide adequate confidence intervals for the hybrid FF $CO_2$ emission estimates. To this end, we had to ensure that potential errors in our top-down estimates of $NO_x$ and CO emissions are statistically independent from those of the conversion factors. We also had to ensure that the evaluation of confidence intervals does not involve any subjective quantitative assumptions regarding the level of uncertainties in measured and simulated data. Such requirements would be difficult to satisfy if the top-down estimates of the emission annual budgets were partly constrained (in the Bayesian sense) with a priori knowledge on these budgets from a bottom-up emission inventory (as it is usual in inverse modeling studies). Furthermore, the use of a priori constraints would make the cross-validation of the estimates of FF $CO_2$ emission budgets based on $NO_x$ and CO measurements infeasible, as both priors and cross validation estimates could then be biased in a similar way due to possible systematic uncertainties in activity data employed in the emission inventory. Accordingly, a distinctive feature of our method is that it does not involve any formal a priori constraints to the top-down estimates of the emission budgets, nor any quantitative settings specifying the level of uncertainties in measured and simulated data. This feature is expected to reinforce the potential of the method to elucidate possible uncertainties and/or inconsistencies in $CO_2$, $NO_x$ and CO emission data provided by different bottom-up emission inventories.

In this study, our method is applied for estimation of annual FF $CO_2$ emissions from a group of 12 western European countries, including a selection of 11 member states of the European Union (EU) that provide the predominant part (>70%) of EU total FF carbon emissions (Ciais et al., 2010b) and Switzerland. Taking into account availability of bottom-up emission inventory data necessary for our analysis, the annual emission estimates were obtained specifically for the year 2008. We believe that estimation of FF $CO_2$ emissions from the European region could be considered as a good testing case for our method, taking into account that uncertainties in corresponding emission inventory data for the EU countries with well-developed statistics are relatively low (compared to potential uncertainties in FF $CO_2$ emission data for countries with




less developed statistical infrastructure), although not quite negligible. For example, by comparing data of several international emission inventories, Ciais et al. (2010b) estimated the full uncertainty of the "bottom-up" estimates of the anthropogenic $CO_2$ emissions in the EU countries to be about 19 % but less (~7 %) if possible inconsistencies between types of $CO_2$ sources taken into account in different emission inventories are resolved. Note that the uncertainty in bottom-up FF

$CO_2$ emission estimates is expected to be lower than in corresponding $NO_x$ and CO emission estimates, because of an important role played for emissions of those proxy species by the technology and end-of-pipe measures; at the same time, the ratio of emissions of $CO_2$ and of the proxy species can be less uncertain than the emission themselves if the emission data are subject to a strong common bias caused by uncertainties in fuel consumption statistics.

In the following, Sect. 2 describes data and modeling tools used in our study. Description of our inverse modeling method

and its validation with synthetic "observations" are presented in Sect. 3. The results of its application to the real-world situation are presented and discussed in Sect. 4. Finally, major findings are summarized in Sect. 5.

## 2 Data and model description

### 2.1 Retrievals from satellite measurements

We used the tropospheric $NO_2$ column retrievals from measurements of the Earth's back scattered radiation in visible and

ultraviolet spectral regions by the OMI satellite instrument (Levelt et al., 2006) onboard the NASA EOS Aura spacecraft. The Aura satellite (Schoeberl et al., 2006) is in a sun-synchronous ascending polar orbit with an equator crossing time of 13:30 local solar time (LST) and an orbital period of 99 min. The OMI instrument has a swath width of ~2600 km divided into 60 pixels with a size of 13-26 km.

The retrievals used in this study are provided by KNMI as the DOMINO version 2 data product (Boersma et al., 2011)

through the TEMIS portal (http://www.temis.nl/). The product contains Level 2 data, that is, $NO_2$ columns and relevant geophysical information for each ground pixel observed by the instrument. In this study, only cloud and surface albedo screened data (retrieved for the scenes with the cloud fraction less than 30 % and with the surface albedo less than 0.3) were used. The main steps of the OMI $NO_2$ retrieval algorithm (see Bucsela et al., 2013 for details) include (1) spectral fitting and the slant column density (SCD) estimation by using the Differential Optical Absorption Spectroscopy (DOAS) method, (2)

calculation of the air mass factor (AMF) defined as the ratio of the measured slant column to the vertical column, and (3) separation of the tropospheric and stratospheric parts of the vertical columns. Each step involves different uncertainties that may contribute, to various extents, to the uncertainty of the tropospheric $NO_2$ columns; in particular, the SCD uncertainty is likely to predominate over other uncertainties in remote areas and is of about $0.75 \times 10^{15}$ molec. $cm^{-2}$, while the retrievals for urban areas are mostly affected by the AMF uncertainty, that is of about 20 % for clear sky conditions (Bucsela et al., 2013).

Several studies (e.g., Zhao et al., 2009, Miyazaki et al., 2012, Vinken et al., 2014) found that in spite of the considerable uncertainties, the tropospheric $NO_2$ columns retrieved from the OMI measurements provide useful constraints to anthropogenic $NO_x$ emissions in different regions of the world, including Europe.



We also used the Level-2 retrievals of total CO column amounts from the measurements performed by the Infrared Atmospheric Sounding Interferometer (IASI) on board the METOP-A satellite (Clerbaux et al., 2009). METOP-A has the sun synchronous polar orbit with equator crossing at 21:30 LST for the ascending node. The IASI instrument provides global coverage twice a day (around 9:30 and 21:30 LST) with a swath of about 2×1100 km and a nominal pixel diameter footprint on the ground of 12 km.

The CO column amounts are retrieved from the cloud screened measurements of the spectrum at the 1-0 rotation vibration band centered at 4.7 μm (2128 cm$^{-1}$) by using the Fast Optimal Retrievals on Layers for IASI (FORLI) algorithm (Hurtmans et al., 2012). The FORLI algorithm provides CO partial column amounts (for at most 19 layers) fitting the spectral observations with a priori constraints; the partial columns are combined to yield the total column amounts. The uncertainty of the IASI CO retrievals strongly depends on the geographical location and conditions of the observations (Clerbaux et al., 2009; George et al., 2009; Turquety et al., 2009); it is estimated to be about 10 % under typical conditions (Clerbaux et al., 2009). It should be noted, however, that the capability of the IASI measurements to inform about CO sources depends not only on the accuracy of the CO retrievals, but also on the sensitivity of the spectral observations to the CO concentration in the boundary layer. A convenient way to characterize this sensitivity (which is related to the vertical resolution of the retrieval and depends, in particular, on the difference between the temperatures of the surface and of the atmospheric boundary layer) is to consider the trace of the averaging kernel matrix (Clerbaux et al., 2009); this parameter is called the Degree of Freedom of the Signal (DOFS). Distinguishing between the upper and lower troposphere requires this parameter to be about 2 (George et al., 2009). Taking these considerations into account, we used only those retrievals that were characterized by relatively large DOFS values: similar to Konovalov et al. (2014), the DOFS threshold was set to be 1.7. The available CO retrievals for individual pixels were projected to the 0.5º×0.5º grid of a CTM (see Sect. 2.2) and averaged over each day.

We would like to note that instead of (or together with) the IASI measurements, we had an option of using alternative data from other infrared sounders, such as MOPITT and AIRS. Our decision to choose the IASI measurements was made by taking into account their relatively high sensitivity in the boundary layer (George et al., 2009), as well as previous studies in which the IASI data were successfully employed for constraining CO emissions from different sources (Fortems-Cheiney et al., 2009; Krol et al., 2013; Konovalov et al., 2014). We considered the relatively high sensitivity of the IASI measurements in the lower troposphere as an important advantage especially in the context of the given study involving a mesoscale CTM. Indeed, the upper troposphere CO content simulated with such a CTM is likely to be strongly affected by boundary conditions which are specified by using global CTM simulations and therefore are not dependent on CO emissions used in the regional CTM (see, e.g., a respective discussion in Konovalov et al., 2011). Exploring the potential of the alternative CO data products goes beyond the scope of the given study.





## 2.2 CTM simulations and initial processing of the model output data

In this study, the relationships between $NO_x$ and CO emissions and, respectively, $NO_2$ and CO column amounts are simulated by the CHIMERE CTM. CHIMERE is a three-dimensional Eulerian model designed to simulate air pollution on urban, regional and continental scales; it allows taking into account most important atmospheric processes (such as

anthropogenic, biogenic and fire emissions, gas-phase and heterogeneous chemistry, advection, turbulent diffusion, and deep convection, dry and wet deposition) affecting the atmospheric fate of a number of reactive gaseous and aerosol species (see Menut et al., 2013 and references therein). The model was earlier successfully used in combination with satellite $NO_2$ and CO retrievals in several inverse modeling studies of $NO_x$ and CO emissions (e.g. Konovalov et al., 2006; 2008; 2010; Beresin et al., 2013; Konovalov et al., 2014; Mijling et al., 2012; Ding et al., 2015).

In this study, the CHIMERE model was run with one of the standard domains (called the CONT5 domain) covering a western part of Europe (-13.75 – 25.25º E; 34.75 – 58.25º N) with the horizontal resolution of 0.5º×0.5º. The simulations were performed with 12 non-equidistant layers in the vertical (up to the 200 hPa pressure level); the layers were specified in the hybrid sigma-p coordinates such that the distance between the layers increased with the altitude from ~50 m near the surface until ~2 km in the upper part of the modeled atmosphere. Gas-phase chemical processes were simulated with the

simplified MELCHIOR2 chemical mechanism (Schmidt et al., 2001), and several heterogeneous reactions on the surfaces of aerosol particles were taken into account as described in Menut et al. (2013). Initial and boundary conditions for gases and aerosols were specified using monthly climatological data from LMDz-INCA global model (Folberth et al., 2006). Meteorological data were obtained from the WRF-ARW (v.3.6) model (Skamarock et al., 2005), which was run with a horizontal resolution of 50km×50km and with 30 levels extending in the vertical up to the 50 hPa pressure level for a region

covering the CHIMERE domain and was driven with the NCEP Reanalysis-2 data (National Centers for Environmental Prediction, 2000). The anthropogenic, biogenic and fire emissions of major gaseous and aerosol species were taken into account in our simulations as described in the next section (Sect. 2.3). The model was run with different scenarios (specified below in Sect. 3.2) for the period from 22 December 2007 to December 29, 2008. The spin-up period included the first 10 days of any run, which therefore were withheld from the following analysis.

To enable consistency of our simulations with the satellite data employed in this study, the CHIMERE outputs were processed by taking into account measurement properties. The simulated $NO_2$ and CO vertical profiles corresponding (in time and space) to each pixel containing, respectively, the OMI and IASI measurements (satisfying to the criteria specified in Sect. 2.1) were transformed into tropospheric $NO_2$ columns and total CO columns, $C_m^{no2}$ and $C_m^{co}$, by applying the respective averaging kernels, $\boldsymbol{A^{no2}}$ and $\boldsymbol{A^{co}}$.

Specifically, the simulated $NO_2$ profile was transformed as follows (Eskes and Boersma, 2003):

$$C_m^{no2} = (\boldsymbol{A^{no2}})^T \, \boldsymbol{C_{m(o)}^{no2}}, \tag{1}$$

where $\boldsymbol{C_{m(o)}^{no2}}$ are the original model outputs (partial columns) interpolated to the pressure grid of the averaging columns. A slightly different procedure was used to process the modeled CO partial columns:

$$C_m^{co} = (A^{co})^T\big(C_{m(o)}^{co} - C_a^{co}\big) + I^T C_a^{co}, \tag{2}$$

where $C_a^{co}$ is the a priori CO vertical profile used in the retrieval procedure, and $I$ is the identity vector. The missing components of $C_{m(o)}^{co}$ for altitudes above the upper layer of the CHIMERE CTM were taken to be equal to the respective values from $C_a^{co}$. Note that the transformation providing the total CO columns in accordance to Eq. (2) is a special case of the

more general transformation procedure providing partial CO columns (see Fortems-Cheiney et al., 2009). The model outputs transformed with different averaging kernels but corresponding to the same model grid cell and hour as the observations were averaged. With the satellite data used in this study, each grid cell is provided with observed or modeled data for at most two different hours of each day. In addition to the selection criterion based on the DOFS values (see Sect. 2.1), in order to minimize the impact of model errors that are not associated with uncertainties in emission data on inverse modeling results,

only those days and grid cells were taken into account when and where the modeled contribution of anthropogenic NO$_x$ or CO emissions in the study region (specified in the next section) to $C_m^{no2}$ and $C_m^{co}$ was larger than one percent of the corresponding "background" values of the columns (here "background" is defined according to a simulation made without anthropogenic emissions, i.e. with the biogenic and open biomass burning emissions specified in the next section, and with the transport model boundary conditions described above).

**2.3 Emission inventory data**

We used annual anthropogenic emission data from several sources: the European Monitoring and Evaluation Programme (EMEP) regional emission inventory (e.g., Bieser et al., 2011), the Emission Database for Global Atmospheric Research, version 4.2 (EDGAR v4.2) (EC-JRC/PBL, 2011), the Carbon Dioxide Information Analysis Center (CDIAC) (Boden et al., 2011). Specifically, the EMEP anthropogenic annual emission data distributed among 11 Selected Nomenclature for Air

Pollutants (SNAP) sectors cover NO$_x$ and CO as well as non-methane hydrocarbons (NMHC), SO$_x$, and particulate matter for the year 2008 on a grid with the resolution of 0.5°×0.5°. The EMEP inventory was used in our simulations described in Sect. 2.2. In addition, the EMEP data for the national totals of NO$_x$ and CO emissions were used in the analysis of uncertainties in our emission estimates (see Sect. 3.4) along with the EDGAR v4.2 and CDIAC data. Note that the EMEP inventory does not provide data for CO$_2$ emissions and that the emissions for the 11[th] sector (comprising biogenic sources

and fires associated with human activities) were replaced in our simulations with data of dedicated inventories (as described in this section below). The EDGAR v4.2 data for the national totals of anthropogenic NO$_x$, CO, and CO$_2$ emissions distributed between several emission sectors (not necessarily coinciding with the SNAP sectors) were used in our inversion procedure (see Sect. 3.2) as the source of information on the relationships between CO, NO$_x$, and CO$_2$ emissions. Note that we used the EDGAR v4.2 FF CO$_2$ emission data excluding CO$_2$ emissions from biofuel burning (that is, the data used were

calculated after "excluding short-cycle organic carbon from biomass burning"), while the corresponding CO and NO$_x$ data included emissions from both fossil fuel and biofuel burning. The CDIAC database used in this study reports only national totals of FF CO$_2$ emissions without sectorial breakdowns and was used in this study for evaluation of uncertainties in our





results. Note that $CO_2$ emissions from cement production have been reported in CDIAC (as well as in EDGAR v4.2) separately from FF $CO_2$ emissions and are not considered in this study.

To facilitate our analysis, the anthropogenic emissions were aggregated into two categories. The first category ("EHI") included the emissions associated mostly with energy and heat production and heavy industries. The second category

("TCO") comprised transport, chemical industry, and all other anthropogenic sources. In the EMEP inventory, the EHI category was defined by aggregating the sources corresponding to the first, second and third sectors of SNAP (combustion in energy and transformation industries, non-industrial combustion plants and combustion in manufacturing industry, respectively). The sectors "1A1a-c" (public electricity and heat production; other energy industries), "1A2" (manufacturing industries and construction) and "1A4" (fuel combustion in residential and other sectors) were allocated into the same

category in the case of the EDGAR inventory. The TCO category aggregated all other anthropogenic sources considered in the EMEP or EDGAR v4.2 inventories.

While specifying these categories, we tried to ensure that, on the one hand, the emissions corresponding to the different categories had distinct spatial distributions (such as, e.g., the emissions from power plants and from transport), and, on the other hand, that the amounts of annual emissions from each category were of the same order of magnitude. By doing so, we

attempt to limit the generation of aggregation errors (Kaminski et al., 2001) in our top-down $NO_x$ and CO emission estimates. However, splitting emission sources between the two categories specified above is rather arbitrary: in this study, we did no attempt analyzing the impact of the source categories definitions on the uncertainty of our emission estimates.

Figure 1 shows the CONT5 domain (employed in this study) of the CHIMERE CTM along with the spatial distributions of total annual anthropogenic $NO_x$ and CO emissions from the selection of 12 western European countries considered in our

analysis according to the EMEP inventory for 2008; it also shows the fractions of the two source categories introduced above. Note that emissions outside of the selected countries (including ship emissions) are not indicated (the corresponding territories are left blank), such emissions constitute minor parts of the total $NO_x$ and CO emissions in the whole model domain shown in Fig. 1 (41% and 30%, respectively, according to the EMEP inventory for 2008). The territory of the United Kingdom is not fully represented in the model domain; however, the emissions from the missing northern part of this country

are rather negligible (~ 0.53% of the total emissions in UK).

It is noteworthy that not only the total emissions (see Fig. 1a, b) but also the fractions of the different emission source categories (see Fig. 1c-f) exhibit considerable spatial variations. The spatial variability of the source category fractions indicates that, given sufficiently accurate observations, an appropriate inverse modeling procedure together with the dense spatial sampling of the atmosphere by satellites may have a potential to distinguish between emissions coming from the

different sources. It can also be noted that the fractions of the same source categories of the $NO_x$ and CO emissions considerably differ (cf. Fig. 1c, d and Fig. 1e, f). In particular, while the $NO_x$ emissions mostly come from the TCO sources, the CO emissions are distributed between the TCO and EHI sources much more evenly. This observation indicates that the measurements of these two proxy species might provide complementary (to a certain extent) information on human activities associated with $CO_2$ emissions, even if atmospheric fates the CO and $NO_x$ emissions were identical.



The annual anthropogenic emission data were distributed at shorter time scales by applying monthly, daily and hourly factors from the standard emission interface of the CHIMERE CTM (Menut et al., 2013); the factors were provided for specific pollutants, the SNAP sectors and countries by IER, University of Stuttgart (GENEMIS, 1994). The seasonal variations specified in this way for the two categories of anthropogenic emissions are shown in Fig. 2. In addition, emissions were

vertically distributed within 1 km by using the profiles (specific for each SNAP sector) provided in the emission interface of CHIMERE. Note that the vertical profiles did not explicitly account for aircraft emissions, which are also included in the EMEP inventory, but are likely to provide a very small contribution (less than 2 %) to anthropogenic $NO_x$ and CO emission in Europe (Tarassón et al., 2004).

Along with the anthropogenic emissions, our model included biogenic emissions (in particular, $NO_x$ emissions from soils

and emissions of isoprene and some other hydrocarbons from vegetation) and emissions of gaseous species ($NO_x$, CO and non-methane hydrocarbons) from open biomass burning (fires). Biogenic emissions were calculated for each grid cell, day and hour within the CHIMERE model by using the European inventory of soil NO emissions (Stohl et al., 1996) and the emission factors and parameterizations from the MEGAN (Model of Emissions of Gases and Aerosols from Nature) model (Guenther et al., 2006). The fire emissions were specified using the daily data provided by the Global Fire Assimilation

System, version 1.0 (GFAS v1.0) fire emission inventory (Kaiser et al., 2012). The fire emissions were distributed in the vertical uniformly up to the altitude of 1 km (similar to Konovalov et al., 2011). Note that according to the data of the GFAS v1.0 and EMEP emission inventories, the total emissions of both $NO_x$ and CO from fires in the countries considered (mainly, in Portugal) in 2008 were rather small (~0.5 % and ~5 % relative to the corresponding FF emission estimates given by the EMEP inventory).

**2.4 Preliminary comparative analysis of the measurement and simulation data**

In this section, we compare the measurement and simulated data and assess to what extent the variability of the $NO_2$ and CO columns over the study region is affected by direct anthropogenic emissions in the same region. Figure 3 shows time series of the daily values of $NO_2$ and CO columns averaged over the study region (see Fig. 1). The model was run both with and without anthropogenic emissions in the study region, and the model results are presented in Fig. 3 after compensating for a

systematic difference with the measurements. Note that the modeled $NO_2$ and CO columns shown in Fig. 3 were processed using averaging kernels (see Eq. 2); a very small difference between the CO columns calculated with and without anthropogenic emissions in the study region partly reflects the relatively low sensitivity of the CO retrievals in the boundary layer (compared to the upper troposphere). The systematic difference (the bias) was evaluated as the average difference between the model data (obtained by running CHIMERE with full emissions) and the corresponding measurements. The

averaging was carried out either directly for the whole annual period considered (see Fig. 3a, b), or for each month independently (see Fig 3c, d).

It can be seen that both the $NO_2$ and CO measurements exhibit strong day-to-day variability. A part of the observed variability is captured by the model, but the amplitude of the variations is typically smaller in the simulations than in the





measurements. Exact reasons for the stronger day-to-day variations in the measurements are not known: one possible reason is that the variations in the measurements may reflect random errors in the retrieval procedures (see Sect. 2.1), while another possible reason is that a part of the variations in the measurements may be due to factors which are not taken into account in our model (such as, e.g., daily variability in the boundary conditions). Apart from the day-to-day variations, both the $NO_2$

and CO columns manifest slower variations. Such variations have a seasonal component in both the measured and simulated $NO_2$ columns, with larger values observed in winter than in summer. A regular seasonal variability is visible also in the simulated CO data; however, similar variability in the corresponding measurement data appears to be offset by slower (probably, inter-annual) variability, which is not reflected in the boundary conditions of CHIMERE. The differences between the measurements and simulations vary from month to month, thus indicating the importance of evaluating the biases on

shorter than annual time scales; this observation is taken into account in our inversion procedure described in Sect. 3.2. It should be noted that a part of monthly biases may be due to errors in the seasonal cycles of the emissions specified in CHIMERE. Figure 3 also shows that while the anthropogenic emissions in the study region provide the predominant contribution to the $NO_2$ columns over the same region, the respective signal in the CO columns is very small.

Figure 4 presents the spatial distributions of the annually averaged $NO_2$ and CO columns derived from OMI and IASI

measurements and simulated with the CHIMERE CTM. Note that only the data taken into account in our analysis are shown. $NO_2$ columns from both the measurements and simulations show very strong spatial variability correlating with the spatial distribution of $NO_x$ emissions (cf. Fig. 4a, c and Fig. 1a); this observation is coherent with findings of earlier studies (e.g. Konovalov et al., 2006; Napelenok et al., 2008; Mijling et al., 2012) demonstrating that satellite retrievals of $NO_2$ columns combined with CTM outputs can provide useful information on the spatial distribution of $NO_x$ emissions on a regional and

even local (e.g. cities) scale. However, the simulations do not reproduce the spatial variability of $NO_2$ columns perfectly. In particular, the $NO_2$ column amounts over the hot spots located in the heavily industrialized Po Valley in Northern Italy, as well as over an industrialized region in the North-Western Germany and Madrid are considerably smaller in the simulations than in the measurements; on the other hand, the simulated $NO_2$ column amounts tend to be larger than the satellite retrievals over Great Britain. These differences may be due to uncertainties in the spatial distribution of $NO_x$ emissions, as well as due

to measurement and simulation errors.

Consistent with the results shown in Fig. 3, the signal from anthropogenic emissions appears to be rather weak and "smeared" in the spatial distribution of the CO columns. There are also big differences between the retrievals and simulations in some locations. Both the retrieved and simulated CO column amounts tend to be elevated over areas where the anthropogenic emissions are particularly large (such as those in Belgium, Germany, England or the Po valley in Italy).

However, Fig. 4f showing the CO columns simulated without anthropogenic CO emissions in the study region and transformed using averaging kernels (see Eq. 2) bears evidence that an "anthropogenic signal" in the spatial variations of the measured CO columns may come mostly from the a priori CO columns employed in the retrieval procedure. Therefore, the preliminary analysis presented in this section indicates that the $NO_2$ measurements can potentially provide much stronger constraints for anthropogenic emissions on a regional scale, compared to the CO measurements.





## 3. Method

### 3.1 Preliminary remarks

Our method is first described below for a rather general case (with arbitrary numbers of proxy species and emission source categories and for an arbitrary region); some settings specific for this study are either explained later or have been discussed

in Sect. 2. The main steps of the method were briefly outlined in Introduction. The key step of the method – namely, the estimation of annual emissions of a proxy species from different categories of sources (emission sectors) in a region of interest – is described in Sect. 3.2. This step involves optimization of the emissions for a given sector by fitting simulations performed with a chemistry transport model (CTM) to satellite observations of a corresponding species. An important element of the first step is the estimation and elimination of a possible systematic discrepancy between the simulations and

observations, which is not related to uncertainties in a priori emission data. Further steps leading to the estimation of the budgets of FF $CO_2$ emissions are described in Sect. 3.3. An important part of the method is dedicated to the estimation of the confidence intervals for all our emission estimates (see Sect. 3.4).

### 3.2 Optimization of emissions of proxy species

We estimate total annual emissions, $E_c^s$, of a given proxy species, $s$, from different source categories, $c$ ($c \in [1, N_c]$, where $N_c$

is the total number of categories) in a study region. To do that, we combine observations, $\boldsymbol{C}_o^s$, of the species atmospheric column amounts with respective modeled data, $\boldsymbol{C}_m^s$, by assuming (similar to e.g. Berezin et al., 2013) that $\boldsymbol{C}_m^s$ depends on the emissions of a corresponding species linearly:

$$\boldsymbol{C}_m^s \cong \boldsymbol{C}_{mb}^s + \sum_c \mathbf{S}_c^s \boldsymbol{a}_c^s (E_c^s - \tilde{E}_c^s),  \tag{3}$$

where $\tilde{E}_c^s$ are the available (a priori) bottom-up annual anthropogenic emission estimates for a species $s$ and a source

category $c$, $\boldsymbol{a}_c^s$ is the vector specifying allocation of the annual anthropogenic emissions to each cell of model's grid and each day of the simulations, $\mathbf{S}_c^s$ is the Jacobean matrix containing sensitivities of the model outputs to the emissions, $\boldsymbol{C}_{mb}^s$ are the species amounts calculated in a "base" model run using the bottom-up emission inventory data.

The annual emission estimates for individual source categories, $E_c^s$, constitute the control vector of our inverse problem, $\boldsymbol{E}^s$. The optimum estimate of $\boldsymbol{E}^s$ can be obtained by minimizing the sum of the squared differences between the observations and

simulations as follows:

$$\widehat{\boldsymbol{E}}^s = agrmin\{(\boldsymbol{C}_o^s - \boldsymbol{C}_m^s + \boldsymbol{\Delta}^s)^T (\boldsymbol{C}_o^s - \boldsymbol{C}_m^s + \boldsymbol{\Delta}^s)\},  \tag{4}$$

where $\widehat{\boldsymbol{E}}^s$ is the optimal estimate of the control vector, $\boldsymbol{\Delta}^s$ denotes the systematic discrepancies between the simulations and observations of a given proxy species $s$. Note that different components of the vectors $\boldsymbol{C}_o^s$, $\boldsymbol{C}_m^s$, and $\boldsymbol{C}_{mb}^s$ are assumed to represent available values of the respective columns amounts of the species $s$ in different grid cells and / or different

moments of time in the region and period considered.

The estimation given by Eq. (4) formally implies that the errors are homoscedastic, normally distributed and uncorrelated in space and time; deviations of real data from these ideal assumptions can result in errors in $\widehat{\boldsymbol{E}}^s$, but we attempt taking such



errors into account in respective confidence intervals for $\widehat{E}^s$ (see Sect. 3.4). The systematic discrepancies $\Delta^s$, that are assumed to be independent of emission uncertainties and are estimated as explained below, can, in principle, be due to systematic errors both in the simulations and observations. For definiteness, $\Delta^s$ is assumed in this study to be due to biases in the simulations; the vector $\Delta^s$ is referred to below as simply "the bias". Formally, it can be defined as follows:

$$\Delta^s = \langle C_m^s - C_o^s \rangle, \tag{5}$$

where the brackets denote the averaging over the assumed statistical ensemble of probable values of $C_m^s$ and $C_o^s$ in a situation when the anthropogenic emissions in the study regions are known exactly.

Note that Eq. (4) does not include any formal a priori constraints on the magnitude of the optimal emission estimates (unlike many other inverse modeling studies), and, accordingly, our procedure does not involve any explicit quantitative settings for the a priori error covariance matrices. In this way, we avoid possible uncertainties in optimal emission estimates that could be associated with such settings. Not using a priori constraints on the magnitude of the optimal emission estimates also enhances the value of the $CO_2$ emission estimates derived from completely independent measurements of different proxy species for cross-validation purposes, because otherwise the top-down estimates of emissions of the proxy species (and, accordingly, hybrid estimates of $CO_2$ emissions) could be more strongly dependent on the data of bottom-up inventories providing a priori estimates. Avoiding formal a priori constraints is feasible as long as the dimension of the control vector is much smaller than that of the measurement vector. It is so in our case because we do not attempt improving the allocation of the emissions in space and time: the vectors $a_c^s$ are assumed to be known (in practice, $a_c^s$ are provided implicitly by an emission interface in a CTM). Similar assumptions are not unusual in inverse modeling studies involving chemistry transport models (e.g., Pétron et al., 2004; Müller and Stavrakou, 2005; Huneus et al., 2012), when the emissions are corrected for big regions rather than for each model grid cell individually: indeed, optimization of emissions of chemically reactive species (like $NO_x$) is, in a general case, a time consuming task, even when an adjoint code is available. A drawback of fixing spatial and temporal distributions of the emissions in inversion is a probable aggregation error (Kaminski et al., 2001), that we attempted to take into account in the uncertainty analysis as explained in Sec. 3.4.

We assess the bias as the difference between the monthly mean values of the simulations and observations:

$$\Delta_i^s \cong \left[ \sum_j \theta_j^s(m) \right]^{-1} \sum_j \left[ \theta_j^s(m)\left(C_{mj}^s - C_{oj}^s\right) \right], \tag{6}$$

$$\begin{cases} \theta_j^s = 1, \ j \in \Omega_m \\ \theta_j^s = 0, \ j \notin \Omega_m \end{cases},$$

where $\Omega_m$ denotes the subset of the available data for a given month $m$, and $i \in \Omega_m$ is the index of a component (a point in time and space) of the vector $\Delta^s$. It should be noted that values of $C_m^s$ in Eq. (6), like those in Eq. (4), depend on the control vector, $E^s$. When combined with Eqs. (3) and (6), Eq. (4) specifies a linear optimization problem that can be easily resolved numerically. Effectively, information about optimal values of the emission vector is obtained from spatial and temporal variations of the observations and simulations within each month.



Eq. (6) provides a simple approximation for Eq. (5) by implying that the systematic differences between different pairs of simulations and observations corresponding to a given month are about the same; that is, we assume that the bias is uniform in space and time during a given month. In reality, however, systematic errors of satellite retrievals and model results can be different for different grid cells and days. Therefore, this approximation (that reflects the lack of any "a priori" information

about the bias) may introduce some extra errors in our emission estimates, which would not appear if the structure of the bias was known exactly. Although we cannot avoid such errors, we try, at least, to take them into account in the confidence intervals for our estimates. Note that as long as there is only one realization of $C_m^s$ and $C_o^s$ for the region and period considered, an unambiguous separation between their random uncertainties and systematic errors is hardly feasible anyway.

Summing up the optimal emission estimates for the different source categories provides the estimate of total emissions,

$\hat{E}_{sum}^s$, of the species $s$ in the study region. Alternatively, the estimate of the total emissions can be obtained by applying the estimation procedure described above to the special case where all emission sources are aggregated together and $N_c = 1$. The corresponding optimal emission estimates are denoted below as $\hat{E}_{tot}^s$. Considering the difference between $\hat{E}_{sum}^s$ and $\hat{E}_{tot}^s$ provides a useful test for self-consistency of the inversion procedure: the difference should not exceed the combined confidence intervals (that are expected to include an aggregation error among other uncertainties) for $\hat{E}_{sum}^s$ and $\hat{E}_{tot}^s$.

In this study, the product of the Jacobean matrix, $\mathbf{S}_c^s$, and of the vector $\boldsymbol{a}_c^s$ (see Eq. 3) was evaluated as the difference between the results of a model "base" run performed with the standard emission settings as described in Sect. 2.3 and the results of the special runs ("EHI" or "TCO") performed after decreasing the annual EMEP emission values for the respective (EHI or TCO) source categories by 10%. The product of $\mathbf{S}_c^s$ and $\boldsymbol{a}_c^s$ in the case where all emission sources were aggregated together (that is, with $N_c = 1$) was evaluated as the sum of the products of $\mathbf{S}_c^s$ and $\boldsymbol{a}_c^s$ for the two individual (EHI and TCO)

emission categories.

### 3.3 Estimation of FF CO₂ emissions

Following Berezin et al. (2013), we introduce the conversion factors, $F_c^s$, describing the relationships between the annual emissions of a given proxy species $s$ and those of CO₂:

$$F_c^s = \frac{\tilde{E}_c^{CO2}}{\tilde{E}_c^s},$$

(7)

where $\tilde{E}_c^{CO2}$ and $\tilde{E}_c^s$ are the annual estimates of anthropogenic emissions of CO₂ and of a species $s$ for a given emission source category (sector) $c$. Here (as above), the tildes indicate that the emission estimates are obtained from a bottom-up emission inventory as opposed to the optimal emission estimates, $\hat{E}_c^s$, inferred from the measurements according to Eq. (4). Application of the conversion factors to the corresponding optimal emission estimates allows us to obtain the "hybrid" CO₂ emission estimates, $E_{sc}^{CO2}$, that are partly constrained by the measurements but also depend on data of the emission inventory:

$$\hat{E}_{sc}^{co2} = F_c^s \hat{E}_c^s.$$

(8)

Similarly, we can estimate the total CO₂ emissions:

$$\hat{E}_{s,sum}^{co2} = \sum_c F_c^s \hat{E}_c^s.$$

(9)



The alternative total $CO_2$ emission estimate, $\hat{E}_{c,tot}^{co2}$, can be inferred directly from an estimate of the total emissions of a proxy species:

$$\hat{E}_{s,tot}^{co2} = F_{tot}^{s} \hat{E}_{tot}^{s}, \tag{10}$$

where $F_{tot}^{s}$ is evaluated similar to Eq. (7) but by using inventory data for the total emissions of $CO_2$ and the proxy species.

The hybrid $CO_2$ emission estimates derived from measurements of different species can be used for the cross-validation purposes (specifically, the different estimates are expected to agree within the range of their confidence intervals if all uncertainties including aggregation errors are adequately accounted for in the inversion procedure). They can also be combined by taking into account the uncertainty of the individual estimates. Specifically, given $N_s$ individual emission estimates, $\hat{E}_{sc}^{co2}$, the combined (maximum likelihood) estimate of the $CO_2$ emissions, $E_{comb,c}^{CO2}$, and its uncertainty range can

be expressed as follows:

$$\hat{E}_{comb,c}^{CO2} = \sum_{s=1}^{Ns} (\sigma_{sc}^{CO2})^{-2} \sum_{s=1}^{Ns} \hat{E}_{sc}^{CO2} (\sigma_{sc}^{CO2})^{-2};$$

$$\sigma_{comb,c}^{CO2} = 1/\sum_{s=1}^{Ns} (\sigma_{sc}^{CO2})^{-2}, \tag{11}$$

where $\sigma_{sc}^{CO2}$ are the uncertainties (the standard deviations) of $\hat{E}_{sc}^{co2}$.

A combined estimate for the total $CO_2$ emissions, $\hat{E}_{comb,tot}^{CO2}$, can be obtained in a similar way by using values of $\hat{E}_{s,tot}^{co2}$. An

alternative combined estimate for the total emissions, $\hat{E}_{comb,sum}^{CO2}$, can be obtained by summing up values of $\hat{E}_{comb,c}^{CO2}$ for different source categories $c$. The standard deviations $\sigma_{sc}^{CO2}$ can be evaluated as described in the next section (Sect. 3.4). Importantly, according to Eq. (11), the probable uncertainty of the combined estimate $E_{comb,c}^{CO2}$ is smaller than the uncertainty of any of the individual estimates. It should be noted, however, that Eq. (11) provides the maximum likelihood estimate only if the "input" emission estimates derived from measurements of individual proxy species are statistically independent from

each other; otherwise it would be necessary to take into account their error covariances. Applicability of Eq. (11) to the situation addressed in this study is discussed in Sect. 4.2.

### 3.4 Uncertainties in the emission estimates

Evaluation of credible confidence intervals for our optimal emission estimates by using a typical error propagation technique requires proper knowledge of the statistical characteristics of model and measurement errors. However, in case of

simulations and satellite measurements of minor atmospheric species, such knowledge is usually lacking due to complexity and multiplicity of factors that may lead to retrieval and model errors. Taking such considerations into account, instead of using the error propagation technique, we follow the so-called subsampling approach (Politis et al., 1999). Subsampling suggests estimating the confidence interval of a sample statistic (e.g., the variance) by considering variability of that statistic among subsamples drawn from the original sample without replacement.

To adopt the subsampling approach in this study, the original set (sample) of input data for a given proxy species $s$ is divided into $n_d$ subsets (subsamples). From each subset, a "partial" independent emission estimate, $\hat{E}_{c,i}^{s}$ ($i \in [1, n_d]$) is inferred. The



partial estimates can be used to evaluate the standard error, $\sigma_c^s$, of $\hat{E}_c^s$ (that is, the standard error of the sample estimate) as follows:

$$\sigma_c^s \cong \sqrt{\frac{1}{n_{d(n_d-1)}} \sum_{i=1}^{nd} \left( \hat{E}_{c,i}^s - \hat{E}_{c(\bullet)}^s \right)^2}, \tag{12}$$

where ($\bullet$) denotes the mean over all the partial estimates. Importantly, the estimation given by Eq. (12) requires the partial

estimates to be statistically independent. If this condition is satisfied, the partial estimates, $\hat{E}_{c,i}^s$, that are involved in Eq. (12), can be regarded as independent "observations" of the same characteristic: deviations between $\hat{E}_{c,i}^s$ and $\hat{E}_{c(\bullet)}^s$ can only be due to errors in the simulated and measured data. In this sense, Eq. (12) essentially evaluates the standard deviation of the mean of individual "observations" (individual top-down emission estimates in our case) affected by random errors. Note that a simple and robust estimation technique involving Eq. (12) is basically the same as one of the oldest and popular techniques

within the subsampling approach, known as replicated sampling (Deming, 1960; Lee and Forthofer, 2006). The standard errors in our estimates, $\hat{E}_{sum}^s$ and $\hat{E}_{tot}^s$, for the total emissions of proxy species can be evaluated in the same way (that is, by substituting $\hat{E}_{sum}^s$ and $\hat{E}_{sum(\bullet)}^s$ or $\hat{E}_{tot}^s$ and $\hat{E}_{tot(\bullet)}^s$ into Eq. (12) instead of $\hat{E}_{c,i}^s$ and $\hat{E}_{c(\bullet)}^s$).

The statistical independence of the partial estimates could not be ensured in our case if different subsets were selected in a quite arbitrary way. The reason is that the model and observation errors tend to covariate both in space and time (as

confirmed by our analysis discussed below in Sect. 3.5). So, on the one hand, the data included in different subsets should be sufficiently separated in time and / or space to avoid co-variation of errors of different partial estimates. On the other hand, the number of the subsets should not be too small to ensure that the standard error estimate is sufficiently reliable (note that statistical inference defined by Eq. (12) is based on $n_d - 1$ degrees of freedom). It was also necessary to take into account that the error structure in temporal and spatial domains can be different.

In view of these considerations, we opted to divide the original dataset into 4 subsets in the temporal domain and 4 subsets in the spatial domain. Each of the subsets in the temporal domain included data for only one season but for the full spatial domain. The spatial subsets were defined as shown in Fig. 4(e, f); each of the subsets included the data for the whole year. The standard error was estimated in accordance to Eq. (12) independently for both "temporal" and "spatial" subsets (that is, $n_d$ was equal 4 in the both cases), and the maximum of the two estimates of $\sigma_c^s$ was selected as the final estimate of the

standard error. Note that such a division allowed us to retain most of the actual error covariances within a given subsample, as the areas and time periods covered by each subset were significantly larger than expected error covariance scales (see Sect. 3.5 for further details). On the other hand, selection of the maximum of the two different $\sigma_c^s$ estimates may result in overestimation of the confidence intervals that can be robustly evaluated by applying t-values (from the Student's distribution with three degrees of freedom in our case) to the standard error estimate.

We expect that apart from random errors in the input data, the error estimate obtained as described above also includes (at least to some extent) the aggregation error (Kaminsky et al., 2001). In this study, that kind of error may be due to aggregation of similar sources in all the countries considered into a single component of the control vector. As contributions





of various sources to the CO and (especially) $NO_2$ columns in the different countries are different, the aggregation error is likely to be manifested as deviations between the different partial estimates. For example, if, in a hypothetical situation, an emission estimate inferred from the full dataset were mostly affected by strong emission sources from only one country, a partial estimate obtained after leaving the measurements over that country out would likely be much less affected by the

same sources, at least in the case of emission estimates of such a short lived species as $NO_x$.

The confidence intervals estimated using Eq. (12) are also likely to account for a part of estimation errors associated with uncertainties in the diurnal and weekly variations of anthropogenic emissions, as well as with uncertainties due to shortcomings in the model representation of chemical processes (including effects of subgrid-scale chemical interactions). Indeed, it seems reasonable to expect that different errors of the emission temporal cycles for different emission sectors and

countries can be manifested as quasi-random deviations between the simulations and measurements in different grid cells and days. Uncertainties in the diurnal variations of emissions are likely to be manifested additionally in the differences between the hybrid $CO_2$ emissions estimates inferred separately from the CO and $NO_x$ measurements, as those measurements are taken in different times of a day (see Sect. 2.1). Uncertainties in simulations of chemical processes and subgrid-scale chemical interactions are likely to have a different impact on the observed $NO_2$ or CO columns in different types of

environments (e.g. rural or urban) and in different seasons; therefore, the respective model errors are likely to differ in different grid cells, and thus such errors are expected to contribute to the emission estimate uncertainties addressed in Eq. (12).

Compared to the diurnal and weekly variations, uncertainties in the seasonal variations of anthropogenic emissions are more probable to result in common systematic biases of $NO_x$ and CO emission estimates. To get an idea about the magnitude of

such biases, we compared the emission estimates for the two cases involving simulations with different seasonal cycles. The first case (referred to below as the "cycle" case) corresponds to the standard seasonal cycles assumed in our model (see Fig. 2). The second ("flat") case corresponds to simulations performed with constant emissions in any month of a year (but yet with the same diurnal and weekly emission temporal profiles as in the "cycle" case). Differences between the emissions estimates obtained for these two cases are likely to strongly exceed the respective uncertainty (because the "flat" case is

evidently unrealistic) and, for this reason, are not formally included in the confidence intervals for our emission estimates.

It should be noted that the qualitative considerations discussed above are by no means intended to strictly prove that the estimations based on Eq. (12) actually account for all possible errors. Nonetheless, taking the above arguments into account and given the fact that both the origins and the statistical characteristics of errors in the measurement, simulation and inventory data involved in our analysis are very poorly known, we believe that the simple and robust subsampling technique

described above provides sufficiently reliable and robust uncertainty estimates and has no serious alternative in the situation considered. Some further arguments supporting reliability of this technique are discussed in Sect. 3.5.

To obtain the confidence intervals for our $CO_2$ emission estimates, we need to combine the uncertainty of our estimates of emissions of proxy species with the uncertainty of the corresponding conversion factors. Ideally, the uncertainty of the conversion factors for source categories that group different sectors (like EHI and TCO) could be obtained e.g. by varying




parameters of a bottom-up inventory (Wang et al., 2013) and provided along with emission data. However, in our knowledge, such information has unfortunately not yet been made available within any inventory except those by Wang et al. (2013) for China. As an alternative way, we suggest that the uncertainty of the conversion factors can be roughly estimated by comparing their values based on data of different emission inventories. In this study, along with the conversion

factors based on the EDGAR v4.2 emission inventory (those values were used to obtain our "best" $CO_2$ emission estimates as described in Sect. 3.3 and denoted in this section simply as $F_c^s$), we considered "alternative" conversion factor values based on the data of other inventories, such as EMEP and CDIAC. The "alternative" conversion factor values are denoted below as $F_c^{s\prime}$. Specifically, we used the EMEP inventory data for $NO_x$ and CO emissions and the CDIAC data for FF $CO_2$ emissions (see Sect. 2.3). As the CDIAC emission data had not been originally distributed among individual emission

sectors, the fractions of the two categories of the $CO_2$ sources were taken to be the same as in the EDGAR v.4.2 inventory. However, only the original CDIAC and EMEP data were used to estimate the conversion factors applied to the total emissions ($F_{tot}^{s}{}^{\prime}$).

Using again the subsampling technique, we roughly estimated the standard error for the conversion factors, $\sigma_{sc}^F$, as follows:

$$\sigma_{sc}^F = \sqrt{\frac{1}{(N_k-1)N_k}\sum_{k=1}^{N_k}\left(F_{c,k}^s - F_{c,k}^s{}^{\prime} - F_{c(\bullet)}^s + F_{c(\bullet)}^s{}^{\prime}\right)^2 + ([F_c^s - F_c^{s\prime}])^2} \ , \qquad (13)$$

where $F_{s,k}^p$ and $F_{s,k}^p{}^{\prime}$ are the conversion factors evaluated individually for each of the 12 countries considered, $c$ is the country index, $N_k$ is the total number of the countries considered ($N_k$ = 12 in this study), and ($\bullet$) denotes the means over the countries. The country scale is used in Eq. (13), because the CDIAC data had not been provided on a spatial grid, and thus we could not consider the same spatial subsamples as those with the data for $NO_2$ and CO columns. The estimations given by Eq. (12) and (13) are based on the same idea, except that unlike Eq. (12), Eq. (13) does not involve the assumption that the

error of a "sample" estimate is completely random in origin; rather, it takes into account that the error may contain both random and systematic components. The latter is evaluated in Eq. (13) as the difference between the estimates $F_c^s$ and $F_c^{s\prime}$ representing the full study region. Actually, that difference may include a part of the random error, so Eq. (13) is likely to overestimate $\sigma_{cs}^F$. Further overestimation may be due to the fact that the differences in Eq. (13) comprise cumulative errors in the both conversion factor estimates: if the errors were distributed equally between the "main" and alternative estimates, a

proper value of $\sigma_{cs}^F$ would be at least the factor of $2^{1/2}$ smaller than the one given by Eq. (13). In contrast, using the same (EDGAR v.4.2) data to evaluate both $F_{c,k}^s$ and $F_{c,k}^s{}^{\prime}$ may compensate such an enhancement or even entail a tendency for underestimation in $\sigma_{sc}^F$ (except for the case where the conversion factors and their uncertainties are estimated directly for total emissions, that is, without sectorial breakdowns). Nonetheless, on the whole, taking the above qualitative considerations into account, we expect that values of $\sigma_{sc}^F$ calculated as described above are more likely to be overestimated than

underestimated, thus being conservative in our approach to provide optimal $CO_2$ emission estimates.

Values of the conversion factors, $F_{tot}^s$ and $F_{tot}^s{}^{\prime}$, calculated using different inventories for each country considered are shown in Fig. 5. The differences between the different estimates of the conversion factors are, in general, considerable and vary across different countries in the study region. Specifically, the differences for the $NO_x$-to-$CO_2$ emission and CO-to-$CO_2$



conversion factors range from 1.4 to 24.9 % and from 3.8 to 52.6 % (relative the values based on the EDGAR v.4.2 data), respectively. The differences are smallest for Austria and Germany.

The standard error, $\sigma_{sc}^{CO2}$, representing the uncertainty in our hybrid estimates of anthropogenic $CO_2$ emissions was estimated by assuming that uncertainties in the estimates of a proxy species emissions and in the estimates of the conversion factors are independent:

$$\sigma_{sc}^{CO2} = \hat{E}_{sc}^{co2} \sqrt{\left(\frac{\sigma_c^s}{\hat{E}_c^s}\right)^2 + \left(\frac{\sigma_{sc}^F}{F_c^s}\right)^2}. \tag{14}$$

The standard error, $\sigma_{s,tot}^{CO2}$, for a corresponding total $CO_2$ emission estimate, $\hat{E}_{s,tot}^{co2}$ (see Eq. 10), was evaluated in the same way. Taking into account that the uncertainties in the top-down estimates of emissions of proxy species for different source categories are likely not independent, the standard error, $\sigma_{s,sum}^{CO2}$, of $\hat{E}_{s,sum}^{co2}$ (see Eq. 9) was given by a similar (although a slightly more complicated) equation:

$$\sigma_{s,sum}^{CO2} = \sqrt{\sum_c \left(\hat{E}_c^s \sigma_{sc}^F\right)^2 + \left(\sigma_{s,sum}^{CO2|F}\right)^2}, \tag{15}$$

where $\sigma_{s,sum}^{CO2|F}$ represents the standard error of $\hat{E}_{s,sum}^{CO2}$ under the condition that the conversion factors are known exactly (that is, the errors included in $\sigma_{s,sum}^{CO2|F}$ are associated with only uncertainties of our top-down emission estimates for the proxy species); $\sigma_{s,sum}^{CO2|F}$ was evaluated by using the same subsampling technique as described above for the case of estimation of uncertainties in $\hat{E}_c^s$. The standard errors given by Eq. (14) or (15) allowed us to combine the estimates based on the measurement of $NO_2$ and CO columns by using Eq. (11).

### 3.5 Observation system simulation experiments (tests with synthetic data)

In this section, we examine the capabilities of our method for estimation of emissions of the proxy species by means of observation system simulation experiments (OSSEs). Specifically, we apply our method to synthetic "observational" data featuring known uncertainties that are evaluated by considering the misfits between real observation and corresponding simulated data. Specifically, to generate the synthetic data, we assumed that the covariances of cumulated errors in the real measurement and simulation data can be described by the three-dimensional covariance function, $cov^s(\rho_x, \rho_y, \rho_t)$, that can be approximated as follows:

$$cov^s(\rho_x, \rho_y, \rho_t) \cong cov_x^s(\rho_x) cov_y^s(\rho_y) cov_t^s(\rho_t), \tag{16}$$

where $\rho_x$ and $\rho_y$ denote the distances between a pair of "observations" in west-to-east and south-to-north directions, respectively, $\rho_t$ is the period (the lag) between different "observations", $cov_x^s$, $cov_y^s$, and $cov_t^s$ are the respective one-dimensional covariance functions. We further approximated the covariance functions by using misfits between the observations and simulations as follows:

$$cov_*^s(\rho_*) \sim \sum_i \sum_j \mathcal{H}_{ij}^s[\rho_*](C_{oi}^s - C_{mi}^s + \Delta_i^s)(C_{oj}^s - C_{mj}^s + \Delta_j^s), \tag{17}$$



where the subscript "*" denotes either $x$, or $y$, or $t$, $\mathcal{H}_{ij}^s[\rho_*]$ is the selection operator which is non-zero (unity) only for those pairs of data points that correspond to a given value of $\rho_*$, $C_{oi}^s$ and $C_{mi}^s$ are the vectors of the observational and simulated data, and $\Delta^s$ is the bias. The distances and the lag were expressed in the numbers of grid cell and days, respectively. The vector $\boldsymbol{C}_o^s$ involved in Eq. (17) represents the actual observational data described in Sect. 2.1. The simulated data, $\boldsymbol{C}_m^s$, were

obtained from the model "base" run results presented in Sect. 2.4, and the bias was evaluated on the monthly basis as the "zero-order" estimate obtained by applying Eq. (6) to the same data (that is, without using top-down emission estimates). The covariance functions, $cov_x^s$, $cov_y^s$, and $cov_t^s$, evaluated according to Eq. (17) were found to have the following characteristic scales (corresponding to a two-fold decrease of the covariance functions): 3 (5) and 2 (3) grid cells and 1 (1) days in the case of $NO_2$ (CO) columns, respectively, although these scales do not necessarily reflect the presence of rather

long "tails" in the covariance functions.

Our OSSEs are not expected to disregard any of possible errors in observational and model data that determine variability of the misfits between the observations and simulations within one month, although it should be noted that Eq. (16) provides a rather simplified temporal and special structure of such errors. In particular, our error model does not allow us to take into account probable error "clusters" that can be associated with the aggregation error in our optimal emission estimates.

Nonetheless, inversion of the synthetic data generated even with the simplified error model is useful, as it allows us to assess the adequacy of our uncertainty estimates obtained with the subsampling technique in the presence of probable covariances of errors in the input data, as well as to examine the self-consistency of our procedure (that is, to see whether or not any systematic deviations of our optimal emission estimates from the "true" emission values are covered by the corresponding confidence intervals).

Estimations given by Eq. (16), (17) were used to set up a Monte Carlo experiment in which the vector $\boldsymbol{C}_m^s$ (either obtained from the model base run or from a model run with the emissions perturbed as explained below) represented the "true" content of a given species, while the synthetic "observations" were generated by adding random errors (and, in some cases, biases) to $\boldsymbol{C}_m^s$. Samples of the errors with the covariance structure given by Eq. (16) were generated from a Gaussian distribution by using the Cholesky decomposition method (Press et al., 1992). Using the synthetic data, we obtained

"uncertain" emission estimates which were compared with "true" emission data specified in the model. Each Monte Carlo experiment included 100 iterations performed with the same covariance matrix and with the same bias, $\Delta^s$, but with different samples of random errors. The bias added to $\boldsymbol{C}_m^s$ in each experiment for any given day was specified by linearly interpolating (in time) the monthly biases shown in Fig. 3(c, d), with the magnitude of the monthly bias values scaled (in different experiments) with a factor ($\delta$) ranging from 0 to 1.

The uncertainties (expressed as the standard error) in the emission estimates were evaluated both in the "direct" way (as the root mean square difference between the "uncertain" and "true" emission estimates) and by averaging squares of $\sigma_c^s$ calculated by using the subsampling technique described above. The magnitude of errors in the synthetic data was changed in different experiments by applying a scaling factor ($\sigma$) ranging from 0 to 1 to the covariance matrix given by Eq. (16). An





additional factor ($\xi$) was introduced to scale the non-diagonal components of the covariance matrix: $\xi$ equals zero in an "ideal" case where errors in each grid cells and days are statistically independent from errors in any other grid cells and days. Along with the experiments where the "true" emissions were set to be exactly the same as the bottom-up emissions used in the base run of our model (see Fig. 6, 7), we performed the experiments where the base case emissions for both $NO_x$ and CO were either uniformly increased by 20 percent (see Fig. 8), or increased by 20 percent only for the EHI categories but reduced by 20 percent for the TCO category (see Fig. 9). Note that only anthropogenic $NO_x$ and CO emissions were perturbed in the corresponding model runs, but also respective anthropogenic emissions of all other model species, including those of NMHCs.

The results of the OSSEs indicate, in particular (see Fig. 6, 7), that if errors in the input data for different grid cells and days were statistically independent ($\xi = 0$), the uncertainties (evaluated in the "direct" way with both $\sigma$ and $\delta$ equal unity) of our top-down estimates of both $NO_x$ and CO emissions would be very small, specifically 0.9 % and 0.6 % for the $NO_x$ emission estimates in the EHI and TCO sectors and somewhat larger (13 % and 5 %) for the CO emission estimates for the same sectors. Expectedly, taking the error covariances into account increases the emission estimate uncertainties considerably. The uncertainties in the estimates of $NO_x$ (CO) emissions from the EHI and TCO sectors are found to be 4 (28) % and 5 (17) %, respectively. Larger uncertainty levels in the CO emission estimates compared to those in the $NO_x$ emission estimates are an expected result reflecting the fact that the constraints to CO emissions provided by the CO observations are much weaker than the corresponding constraints provided by the $NO_2$ observations to the $NO_x$ emissions. Indeed, an "emission signal" in the CO data considered (see Figs. 3&4) is, on average, much weaker than that in the $NO_2$ data; moreover, taking into account that the atmospheric lifetime of CO is much longer compared to that of $NO_x$, an emission signal from a given grid cell is effectively spread between a much larger number of grid cells (and days) in CO- than in $NO_2$ observations, resulting in large non- diagonal elements of the Jacobian matrix and potentially leading to a stronger sensitivity of the CO emission estimates to errors in the input data. Interpretation of changes in the uncertainty estimates with respect to the "ideal" case is difficult: it can only be speculated that the increase in the uncertainties is larger in the $NO_x$- than in CO emission estimates, probably because introduction of the error covariance is effectively equivalent to aggregation of available observations into a few "super-observations", leading to suppression of the effect of large non-diagonal elements in the Jacobian matrix describing the relationship between the CO emissions and observations.

Importantly, it is found that introduction of a bias into the synthetic data does not have a strong impact on the accuracy of the retrieved emission estimates (see Fig. 7) affected by random errors in the input data. This result confirms that our inverse modeling scheme is indeed capable of efficiently filtering out the bias, even if it is not constant during one month (as assumed in Eq. 6).

The results of our OSSEs also indicate that the subsampling technique employed in this study provides reasonable uncertainty estimates, although tends to overestimate the actual uncertainties in the experiments representing the most realistic case (where all the scaling factors equal unity). We consider a probable overestimation of uncertainties in our emission estimates as a rather positive feature of our procedure, making conclusions of this study more reliable.





The results shown in Figs. 8 and 9 demonstrate that the optimal emission estimates obtained with our inversion procedure are likely not significantly biased even if the true emissions are considerably different from the bottom-up emission inventory data. These results also confirm that our inversion procedure enables efficient separation of the uncertainties in the model data due to emission errors from other systematic uncertainties in the model and observation data. Importantly, the fact that

the emissions perturbations are retrieved almost perfectly, indicate that the effects of chemical interactions (nonlinearities) and changes in NMHC emissions on the relationships between $NO_x$ and CO emissions and the $NO_2$ and CO columns, respectively, are likely rather small in the situation considered, although it should be noted that such effects can be stronger if the differences between the bottom-up and true emissions were much larger than in our experiments (± 20 %).

## 4 Results

### 4.1 $NO_x$ and CO emission estimates

The estimates of anthropogenic $NO_x$ and CO emissions from the "EHI" and "TCO" categories ($\widehat{E}_c^s$) as well as from all sources aggregated together ($\widehat{\boldsymbol{E}}_{tot}^s$) based on actual observations are presented in Fig. 10. The corresponding numbers are listed in Table 1 that also shows "alternative" estimates ($\widehat{\boldsymbol{E}}_{sum}^s$) of the total emissions. The results are reported for the two estimation cases ("cycle" and "flat", see Sect. 3.4) that involve different seasonal variations of anthropogenic emissions in

the model (specifically, the seasonal cycles specified in the standard version of the CHIMERE CTM were used for the "cycle" case estimations, while constant monthly anthropogenic emissions but with diurnal and weekly variations were employed for the "flat" case). Note that the "flat" case is obviously unrealistic and is considered here only for testing purposes; accordingly, if not stated otherwise, below we discuss estimates obtained for the main ("cycle") case. The uncertainties are reported in terms of the 68.3 % (1-sigma) confidence intervals: a probable overestimation of uncertainties

by our subsampling technique (see Sects. 3.4 and 3.5) suggests that the uncertainty intervals actually correspond to a higher confidence level.

All of our optimal (top-down) estimates of both the total $NO_x$ and CO emissions are slightly (less than 10 percent) smaller than the bottom-up estimates based on the EMEP inventory data; the differences between the top-down and bottom-up estimates are not statistically significant. The relative uncertainties in our estimates of the total emissions range from 10 %

(in case of the $\widehat{E}_{tot}^s$ estimate for the $NO_x$ emissions) to 30 % (in case of the $\widehat{E}_{sum}^s$ estimate for the CO emissions). A lower uncertainty in our estimate of the total $NO_x$ emissions is not quite surprising, as random uncertainties of a huge amount of individual retrievals used in our inverse modeling analysis tend to compensate each other, while systematic errors were taken into account in the framework of our inversion procedure explicitly. Nonetheless, these low uncertainty estimates should be considered with a certain degree of caution as they may not fully account for some unknown errors depending on emissions

themselves (e.g. due to chemical nonlinearities), even though the results of the OSSEs presented in Sect. 3.5 indicate that such errors are hardly significant in the situation considered. Taking into account our preliminary analysis (see Sect. 2.4) and





the results of the OSSEs (see Sect. 3.5), it is also not surprising that the uncertainties in our CO emission estimates are much larger than those in the $NO_x$ emission estimates.

The differences between our alternative estimates of the total emissions, $\widehat{\boldsymbol{E}}_{sum}^s$ and $\widehat{\boldsymbol{E}}_{tot}^s$, are also small compared to the uncertainties of those estimates, while the uncertainties in $\widehat{\boldsymbol{E}}_{sum}^s$ are larger than the uncertainties in $\widehat{\boldsymbol{E}}_{tot}^s$. The difference

between the uncertainties in $\widehat{\boldsymbol{E}}_{sum}^s$ and $\widehat{\boldsymbol{E}}_{tot}^s$ would be difficult to predict a priori, particularly because the cost function (see Eq. 4) employed in this study includes the bias whose estimation may increase uncertainties in the emission estimates to a various extent, depending on the number of variables to be optimized. Our emission estimates for the individual source categories are much more uncertain than the estimates of the total emissions: the uncertainties range from 15 % in the case of the $NO_x$ emission estimate for the TCO sector up to 54 % in the case of the CO emission estimate for the EHI sector. The

absolute differences of our estimates of both CO and $NO_x$ emissions with the EMEP data are smaller than the respective uncertainty range. It may be noteworthy that our estimates for both the CO and $NO_x$ emissions from the TCO sector are ~12 % lower than the corresponding EMEP estimates. This observation indicates that there may be a common bias in the EMEP data for both $NO_x$ and CO emissions in this sector; however, available information does not allow us to make a firmer conclusion in this regard.

Unlike the EMEP data, the EDGAR v4.2 data strongly disagree with our estimate for the $NO_x$ emissions from the EHI sector. The differences of our estimates with the EDGAR v4.2 data are also larger than with the EMEP data in particular in the cases of the $\widehat{\boldsymbol{E}}_{tot}^s$ estimates of both $NO_x$ and CO emissions and in the case of CO emission estimate for the EHI sector, although smaller in the cases of the $NO_x$ and CO emission estimates in the TCO sectors. In general, our analysis indicates that the $NO_x$ and CO emission data provided by the EMEP inventory are more consistent with the $NO_2$ and CO satellite

measurements than those given by the EDGAR v4.2 inventory. This is an expected result because the methodology used in the EMEP inventory is specific to national statistical data available from European countries, while the EDGAR v4.2 inventory uses a more robust approach applicable at the global scale.

The differences between the estimates obtained with different types of seasonal variations of anthropogenic emissions are small compared to the uncertainty in the estimates for the "cycle" case, although not entirely negligible. Evidently, these

differences cannot explain the significant disagreement of our $NO_x$ emission estimates with the EDGAR v4.2 data. Nonetheless, our test results indicate that the effect of possible inaccuracies in the seasonal variations of emissions may not be negligible and should not be disregarded "a priori" when examining the significance of the differences between the top-down estimates of annual emissions and respective bottom-up inventory data. Note that the uncertainties in the $NO_x$ and CO emission estimates for the individual source categories tend to be larger for the "flat" case than for the "cycle" case, while, on

the contrary, the uncertainties in the total $NO_x$ emission estimates are larger for the "cycle" case. Such non-symmetrical differences indicate that none of the cases considered represent the seasonal cycles in $NO_x$ and CO emissions quite perfectly. The lower uncertainty of the total $NO_x$ emission estimates in the "flat" case does not necessary means that those estimates are more reliable than the ones obtained in the "cycle" case, particularly because, as it in noted in Sect. 3.4, our confidence intervals cannot fully account for probable biases associated with errors in the seasonal cycles of emissions.



The uncertainty levels in our estimates of both $NO_x$ and CO emissions using actual data are considerably larger than those obtained above in our OSSEs (see Sect. 3.5) in which synthetic data were generated using a simplified error model. (Note that to be compared with the confidence intervals discussed in this section, the standard errors presented in Sect. 3.5 should be multiplied with the $t$-score of about 1.2). This result indicates that, as expected, the uncertainties in our emission estimates

are caused not only by random uncertainties in the input (measurement and simulation) data, but also by other factors – such as, e.g., the aggregation error and spatial variability of the bias – which could not be taken into account in our tests adequately. Besides, the actual temporal and spatial structure of both the model and measurement errors is likely much more complex and irregular than that assumed in Eq. (16). Anyway, unless the data subsamples defined in Sect. 3.5 are strongly affected by temporal and spatial covariances of errors in the input data (as evidenced by our OSSEs, that is unlikely the case

in this study), the confidence intervals provided by the subsampling technique are expected to be sufficiently reliable even in such a complex real situation as that considered in this study.

Note that the uncertainties of our top-down estimates of $NO_x$ emissions in the region considered turned out to be comparable with the differences between similar estimates provided by different emissions inventories, or even smaller than them. Therefore, our "top-down" $NO_x$ emission estimates can be considered as an independent alternative to "bottom-up" estimates

based on emission inventory data alone. Both our $NO_x$ and CO emission estimates could formally be combined (in the Bayesian way) with the "bottom-up" (a priori) estimates; the uncertainties in the combined (a posteriori) estimates would probably be lower than the uncertainties in either the top-down or a priori estimates taken alone.

In general, our results confirm the findings of previous studies (see the corresponding references in Introduction) showing that $NO_2$ and CO retrievals from satellite measurements can provide useful information on $NO_x$ and CO emissions over high

emission regions. In this regard, it can be noted that while previous inverse modeling studies utilized satellite CO measurements to estimate CO emissions from regions with predominantly anthropogenic sources involved global CTMs (e.g., Pétron, 2004; Fortems-Cheiney et al., 2009; Kopacz et al., 2010; Jiang et al., 2015), we obtained reasonable top-down CO emission estimates by using a regional model. We regard this fact as a promising development, because the use of regional models (usually featuring a higher spatial resolution than global CTMs and employing high resolution regional

emission inventories that are likely more accurate and detailed compared to global ones) in inverse modeling procedures can, potentially, provide more detailed and accurate constraints to CO emissions from various sources. A major difficulty that needs to be overcome in applications of a regional CTM for estimating anthropogenic CO emissions by inverse modeling is associated with probable biases in boundary conditions, especially for CO which has a long chemical lifetime compared to the transit time across the European domain; here we tackled this difficulty by means of a special procedure aimed at

eliminating systematic differences between the measured and simulated data. The results of the OSSEs presented in Sect. 3.5 (see Figs. 8, 9) indicate that our estimation procedure successfully relies on the spatial gradients of CO (and $NO_2$) columns within the European domain to constrain the CO (and $NO_x$) emissions rather than on the average abundance (which is strongly driven by the boundary conditions) in the measurements.



## 4.2 Fossil fuel $CO_2$ emission estimates

Our hybrid FF $CO_2$ emission estimates presented in this section were obtained by applying the conversion factor values listed in Table 2 to our "top-down" estimates of $NO_x$ and CO emissions discussed above. The FF $CO_2$ emission estimates derived from $NO_2$ and CO measurements ($\hat{E}_{sc}^{co2}$ and $\hat{E}_{s,tot}^{co2}$) as well as the combined FF $CO_2$ emission estimates ($\hat{E}_{comb,c}^{co2}$ and

$\hat{E}_{comb,tot}^{co2}$) are shown in Fig. 11 in comparison with the corresponding data of emission inventories. The same estimates are listed in Table 3, which, in addition, presents another version of the hybrid estimates of total FF $CO_2$ emissions, $\hat{E}_{s,sum}^{co2}$ and $\hat{E}_{comb,sum}^{co2}$ (see Sect. 3.3). Two types of the confidence intervals are provided along with the $CO_2$ emission estimates based on measurements of one proxy species. The "full" confidence intervals include the uncertainty in the top-down estimates of the proxy species as well as the uncertainty in the conversion factors. The "partial" confidence intervals were estimated by

taking into account only the uncertainty in the top-down estimates of the $NO_x$ and CO emissions.

The relative differences of $NO_2$- or CO-measurement-based FF $CO_2$ emission estimates with the EDGAR v4.2 $CO_2$ data replicate the corresponding differences of our top-down $NO_x$- or CO emission estimates with the EDGAR v4.2 data for the respective species. This is not surprising, as the conversion factors relating $CO_2$ emissions with the respective proxy species were based on the EDGAR v4.2 inventory. The "full" relative uncertainties in our $CO_2$ emission estimates are larger than the

uncertainties in our estimates of emissions of proxy species, due to uncertainties in the conversion factors. Among the uncertainties in the conversion factors, $\sigma_{sc}^F$, they are largest for the $NO_x$-to-$CO_2$ and CO-to-$CO_2$ emission conversion factors for the EHI source category (58 % and 38 %, respectively).

These uncertainties strongly contribute to the confidence intervals of the respective $CO_2$ emission estimates. In contrast, the uncertainties are relatively small in the $NO_x$-to-$CO_2$ and CO-to-$CO_2$ emission conversion factors for the total emissions (4%

and 22%, respectively); those uncertainties contribute considerably to the "full" confidence intervals only for the total $CO_2$ emission estimates based on the CO measurements, while the uncertainty of the respective $NO_2$-measurement-based estimate is mostly due to the uncertainty in the top-down $NO_x$ emission estimates. Note that as discussed in Sect. 3.4 and 3.5 our method is likely to overestimate uncertainties in both the top-down estimates and in the conversion factors.

Taking into account the full confidence intervals (which are, in some cases, very wide), all our estimates are in agreement

with the EDGAR v4.2 data, except for the estimates of the total $CO_2$ emissions ($\hat{E}_{s,tot}^{co2}$) based on $NO_2$ measurements and on both $NO_2$ and CO measurements. Our hybrid $NO_2$-measurement-based and combined estimates of the total $CO_2$ emissions (2.67 and 2.71 Pg $CO_2$ with the relative uncertainties of about 10 %) are 12 % and 11 % lower than the EDGAR v4.2 estimate (3.03 Pg $CO_2$), respectively. These differences are statistically significant but at the edge of significance with the given confidence level. Note that while discussing statistical significance of the differences between the hybrid and bottom-

up emission estimates, we do not take into account the uncertainty in the bottom-up inventory data, which has not been reported. The differences between the same hybrid estimates and the corresponding estimate (2.86 Pg $CO_2$) provided by the CDIAC inventory (7% and 5%) are slightly smaller than the differences with the EDGAR v4.2 data and are not statistically significant. Therefore, our analysis suggests that the $CO_2$ emissions in the region considered are likely estimated more





accurately by CDIAC than by EDGAR v4.2: however, the difference between the data of the two inventories in the case considered is small (~6 percent).

Note that if the conversion factor uncertainties were not taken into account, which is not recommended, the difference between our $NO_2$-measurement-based $CO_2$ emission estimate for the EHI sector and the respective EDGAR v4.2 estimate

would be statistically significant. However, it is not significant with respect to the full confidence interval. Considering the emission estimates for the EHI sector along with the total $CO_2$ emission estimates illustrates a possible way of using our method for evaluation of bottom-up FF $CO_2$ emission inventory data. That is, a difference between hybrid and bottom-up estimates that exceeds uncertainties associated with measurement and model errors may, in a general case, be due to the two following reasons: (1) there are inconsistencies between bottom-up estimates of emissions of $CO_2$ and of a corresponding

proxy species or/and (2) a bottom-up $CO_2$ emission estimate is inaccurate. Taking uncertainties in the conversion factors into account allows examining the first reason: evidently, it cannot be ruled out in the case of the emission estimates for the EHI sector. However, the first reason alone is not sufficient to fully explain the differences between the hybrid and bottom-up estimates of the total $CO_2$ emissions.

Comparing $NO_2$- and CO-measurement-based $CO_2$ emission estimates (which, ideally, should be the same) enables their

cross-validation. All kinds of $NO_2$-measurement-based $CO_2$ emission estimates are found to be consistent with the respective CO-measurement-based estimates in the sense that their confidence intervals are intersecting. In principle, this is an important result confirming that uncertainties in our emission estimates are not underestimated, since $NO_2$ and CO measurements are independent from each other. However, it should be noted that the uncertainties in CO-measurement-based estimates are so large that such estimates can hardly be useful as independent source of information on $CO_2$ emissions.

Similar large uncertainties are associated with $NO_2$-measurement-based $CO_2$ emission estimates for the EHI and TCO sectors, as well as with the total $CO_2$ emission estimates obtained by summing the $NO_2$-measurement-based estimates for the individual sectors together. While the uncertainties in the CO-measurement-based estimates are mostly caused by uncertainties in the top-down estimates of CO emissions, the uncertainties in the $NO_2$-measurement-based estimates are mainly associated with uncertainties in the conversion factors.

Importantly, the combined estimates (based on both $NO_2$ and CO measurements) of the FF $CO_2$ emissions from individual sectors feature considerably lower relative uncertainties (evaluated with Eq. 11) than the uncertainties in the estimates based on measurements of only one proxy species (for example, relative uncertainties of 39 % and 42 % for the $NO_2$- and CO-measurement-based estimates, respectively, are reduced to a relative uncertainty of 29 % in the combined estimate for the TCO sector). This fact illustrates the potential usefulness of combining hybrid estimates based on independent measurements

of different proxy species such as $NO_2$ and CO. The uncertainty of our combined estimate ($\hat{E}_{comb,tot}^{co2}$) of the total $CO_2$ emissions is very insignificantly smaller than then the uncertainty of the corresponding $NO_2$-measurement-based estimate.

As mentioned in Sect. 3.4, the uncertainty intervals for our combined estimates evaluated with Eq. (11) can be reliable only if the hybrid emission estimates derived from measurements of individual species are statistically independent. We believe that the $CO_2$ emission estimates derived from $NO_2$ and CO measurements are indeed sufficiently independent particularly





because $NO_2$ (as a part of the $NO_x$ chemical family) and CO experience very different atmospheric processing. Indeed, while the key role in spatial and temporal variations of CO is played by the transport processes (and boundary conditions in simulations), atmospheric evolution of $NO_2$ is very strongly affected by local photochemistry. Thus it seems reasonable to believe that possible model errors for these species are, for the most part, different in origin and weakly correlated. Any

significant covariance of errors in CO and $NO_2$ measurement data is also hardly possible, as those measurements are performed with different satellite instruments and by using different methods (see Sect. 2.1). The covariance of errors in the conversion factors $F_c^s$ for the different species is likely small, too (given the complexity of data involved in bottom-up estimates of different proxy species and the fact that $NO_x$ and CO emissions depend on different technological factors and end-of-pipe measures), although we could not evaluate it confidently with available information. Therefore, the uncertainties

in our combined emission estimates are based on an (so far) inevitable assumption that errors in the conversion factors for the different proxy species are statistically independent.

As in the case with the top-down estimates of $NO_x$ and CO emissions, our hybrid estimates of FF $CO_2$ emissions are rather insensitive to the changes in simulations associated with using different seasonal cycles (cf. the estimates for the "cycle" and "flat" cases). That is, we can conclude that the impact of uncertainties in the assumed seasonal cycles of anthropogenic

emissions on our hybrid estimates is small. In particular, such uncertainties can hardy explain the rather considerable difference between our "combined" estimate of the total $CO_2$ emissions and the corresponding estimate based on the EDGAR v4.2 inventory.

## 5 Summary and conclusions

We examined feasibility of estimation of fossil-fuel (FF) $CO_2$ emissions by using $NO_2$ and CO column retrievals from

satellite measurements. FF $CO_2$ emissions are an important component of the global carbon balance and are believed to be a major contributor to global warming. Although such emissions are usually known with better certainty than $CO_2$ fluxes associated with the biosphere, there still exist considerable divergences between data of different "bottom-up" FF $CO_2$ emission inventories; typically, such data cannot be evaluated by using atmospheric $CO_2$ measurements and rarely come with a reported uncertainty structure.

We followed the concept of "proxy" species that suggests constraining FF $CO_2$ emissions by using atmospheric measurements of minor species co-emitted with $CO_2$. We developed a general inverse modeling method aimed at estimation of the budgets of FF $CO_2$ emissions from different sectors of economy in a given region by using satellite measurements of proxy species. The method involves (1) obtaining "top-down" estimates of anthropogenic emissions of a proxy species from the satellite measurements and simulations performed with a mesoscale chemistry transport model (CTM), (2) using

"bottom-up" emission inventories to relate $CO_2$ emissions with emissions of the proxy species, and (3) combining $CO_2$ emission estimates derived from measurements of different proxy species. Important parts of our method are robust techniques to estimate systematic differences between the measured and simulated data, as well as uncertainties in "top-down" estimates of the proxy species.





The method was applied to a western European region including 12 countries by using the $NO_2$ and CO column amounts retrieved from, respectively, the OMI and IASI satellite measurements along with the simulated data from the CHIMERE CTM. The study region was selected by taking into account that uncertainties in available bottom-up emission inventory data for the EU countries with well-developed statistics are likely rather low, compared to potential uncertainties in FF $CO_2$

emission data for countries with less developed statistical infrastructure, although such uncertainties are likely not quite negligible even in the study region. The relationship between FF $CO_2$ emission and emissions of the proxy species was represented by the $NO_x$-to-$CO_2$ and CO-to-$CO_2$ emission conversion factors evaluated with the EDGAR v4.2 emission inventory. The estimates were obtained for the total FF $CO_2$ emissions from the region considered as well as individually for FF $CO_2$ emissions aggregated into two different source categories (sectors), such that the first category ("EHI") included the

emissions associated mostly with energy and heat production and heavy industries, and the second category ("TCO") comprised transport, chemical industry, and all other anthropogenic sources. Our FF $CO_2$ emission estimates were compared with the corresponding data of the EDGAR v.4.2 global emission inventory; in addition, our total FF $CO_2$ emission estimates for the study region were compared with the data of the CDIAC FF $CO_2$ emission inventory. The top-down estimates of $NO_x$ and CO emissions were compared with the respective data from the European EMEP and global EDGAR v.4.2 emission

inventories.

As expected (taking into account findings of several previous studies), the $NO_2$ column retrievals from OMI measurements provide rather strong constraints to $NO_x$ emissions. Our most reliable "top-down" estimate of the total $NO_x$ emissions is found to be insignificantly (by about 6%) lower than the respective "bottom-up" estimate based on the EMEP emission inventory, while our estimates for the emissions from the "EHI" and "TCO" having much larger uncertainties (of about 18 %

and 15 %, respectively) are found to be in agreement with the corresponding estimates based on the EMEP emission inventory. Larger and statistically significant differences are found between our $NO_x$ emission estimates and the respective data of the EDGAR v4.2 global emission inventory. In particular, our results suggest that the total $NO_x$ emissions from the study region may be overestimated in the EDGAR v4.2 inventory by ~13 %, while the EDGAR emissions for the EHI sector are likely overestimated by more than 60%.

In contrast to the $NO_x$ emission estimates, our top-down estimates of the CO emissions are fully consistent with both the EMEP and EDGAR v4.2 emission data; however, this consistency is partly due to much larger uncertainties in our CO emission estimates (compared to uncertainties in the $NO_x$ emission estimates). The relatively large uncertainties in the top-down CO emission estimates (~55 % and ~35 % in the estimates for the EHI and TCO sectors, respectively, and ~25 % in the total CO emission estimate) are not surprising in view of the much smaller sensitivity of the satellite CO measurements

to anthropogenic CO emissions in the study region, compared to the sensitivity of the $NO_2$ measurements to the anthropogenic $NO_x$ emissions. Nonetheless, in spite of the large uncertainties (which may be overestimated by our procedure), the differences between our top-down estimates of CO emissions and respective EMEP data are rather small (less than 7%). Similar to our $NO_x$ emission estimates, the top-down CO emission estimates differ more considerably from the EDGAR v4.2 data.





The top-down estimates of the $NO_x$ and CO emissions were used to obtain different "hybrid" estimates (combining different information coming from measurements and bottom-up inventories) of $CO_2$ emissions. The $NO_2$-measurement-based hybrid estimate of total $CO_2$ emissions is about 12 % smaller than the respective estimates based on the EDGAR v4.2; the difference exceeds the estimated uncertainty range (~ 11 %) of our estimate, although only marginally. In contrast, the difference between the same hybrid estimate and the corresponding estimate provided by the CDIAC inventory (~ 7 %) is not statistically significant. A large negative difference (more than 60 %) is found between our $NO_2$-measurement-based $CO_2$ emission estimate for the EHI source category and the corresponding EDGAR v4.2 estimate. This difference is, however, not statistically significant and can be mostly attributed to uncertainties in the $NO_x$-to-$CO_2$ emission conversion factor for the given source category. Our CO-measurement-based hybrid estimates of the total FF $CO_2$ emissions are larger than the respective bottom-up estimates based on both the EDGAR v4.2 and CDIAC data but the differences are not too big (less than 25 %) and can be well explained by uncertainties in our estimates. Similar to the case with the $NO_2$-measurement-based hybrid estimate, the largest difference between our CO-measurement-based FF $CO_2$ emission estimates and the EDGAR v4.2 data is found for the EHI source category, with our best estimate being about 26 % larger.

Taking into account the range of uncertainties, all our $NO_2$-measurement-based $CO_2$ emission estimates were found to be consistent with the respective CO-measurement-based estimates. This is an important result confirming the reliability of our approach. The combined emission estimates (based on both $NO_2$ and CO measurements) for individual source categories feature considerably smaller uncertainties than the corresponding "partial" estimates. Our combined estimate of total FF $CO_2$ emissions is weighed toward the $NO_2$-measurement-based estimate and is found to be ~11 % and ~5 % lower than the respective estimates based on the EDGAR v4.2 and CDIAC data. The difference of our estimate with the EDGAR v4.2 data slightly exceeds the confidence interval of our combined estimate, while the difference with the CDIAC data is again not statistically significant.

In general, our analysis demonstrated that $NO_2$ and CO column retrievals from satellite measurements provide reasonable constraints to FF $CO_2$ emissions at the scale of Western Europe. Differences between "hybrid" $CO_2$ emission estimates derived from such data and respective estimates based on bottom-up emission inventory data can, in principle, be due to various kinds of uncertainties in the hybrid estimates (including uncertainties in top-down estimates of emissions of proxy species and uncertainties in the conversion factors). We argue that such uncertainties can be evaluated by using the robust techniques described in this paper. Although relative uncertainties in our top-down CO emission estimates were evaluated to be considerably larger than in the similar $NO_x$ emission estimates based on $NO_2$ measurements, the CO column retrievals were found to be a useful source of independent information on FF $CO_2$ emissions, particularly in the cases where probable uncertainties in the conversion factors for $NO_x$ emissions are larger than uncertainties in the conversion factors for CO emissions. Possible future developments of our approach can, in particular, include (1) using $NO_2$ and CO retrievals from measurements performed by other satellite instruments (such as GOME-2, MOPITT and AIRS) together with the retrievals from the OMI and IASI measurements (as in this study), (2) using an ensemble of different bottom-up emission inventories (when available) to estimate the uncertainty in the conversion factors more accurately, and (3) implementing hybrid $CO_2$



emission estimates into a global transport model simulating $CO_2$ distribution in the atmosphere in order to validate them against ground-based and satellite $CO_2$ measurements. Finally, it should be noted that as FF $CO_2$ emission inventory data for the western European countries are likely much less uncertain than similar data for developing regions of the world, applications of our method to developing regions can be especially fruitful. In this regard, our method can become an

5 integral part of a policy-relevant global carbon observing system (Ciais et al., 2014, 2015).

*Acknowledgements.* The study was supported by the Russian Science Foundation (grant No. 15-17-10024). We acknowledge the free use of tropospheric $NO_2$ column data from the OMI sensor from www.temis.nl and the use of CO column data from http://ether.ipsl.jussieu.fr/ether/pubipsl/iasi_CO_uk.jsp.



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







**Figure 1: Spatial distributions of $NO_x$ (a) and CO (b) total annual emissions (g cm$^{-2}$ yr$^{-1}$) and the fractions (%) of the EHI (c, e) and TCO (d, f) emission source categories (see the definitions in Sect. 2.3) according to the EMEP inventory for 2008. The emission data are shown only for the study region comprising land territories of 12 European countries.**





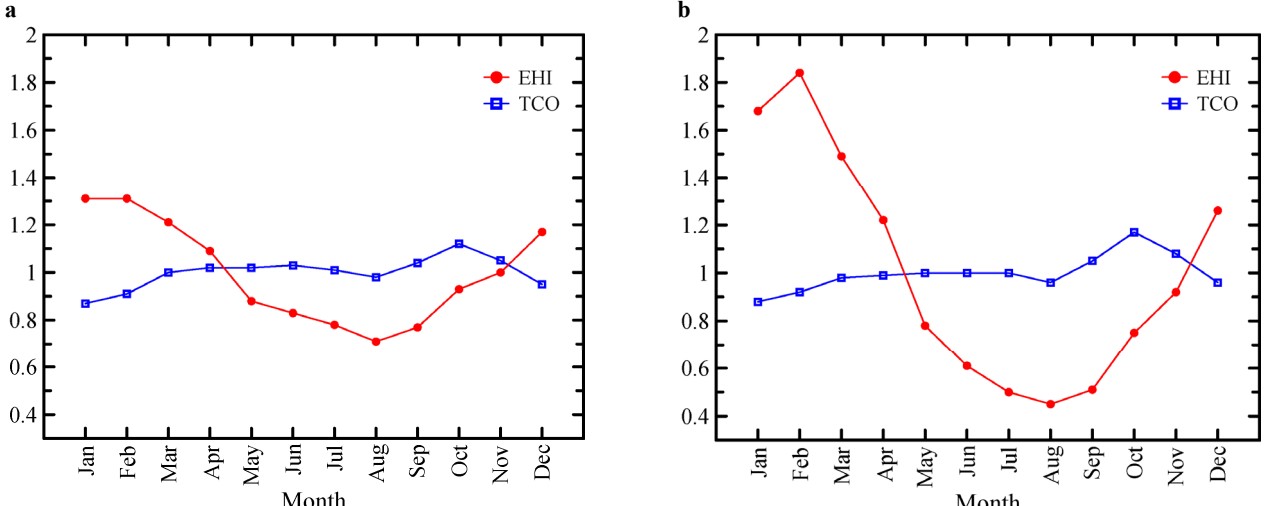

**Figure 2: The seasonal variations of the spatially averaged (over the study region) NO$_x$ (a) and CO (b) emissions for the "EHI" and "TCO" categories of sources. The variations were calculated as explained in Sect. 2.3.**





**Figure 3: Time series of the spatially-averaged NO₂ (a,c) and CO (b,d) columns retrieved from satellite measurements (see green curves) and simulated using the CHIMERE CTM both with and without anthropogenic emissions in the study region (see red and blue curves, respectively). The simulated data shown have been de-biased: the differences (see brown curves) between either the annual (a,b) or monthly (c,d) averages of the simulation and measurement data were subtracted from the original simulation data.**







**Figure 4:** Spatial distributions of the annually averaged $NO_2$ (a, c, e) and CO (b, d, f) columns obtained from satellite observations (a, b) and model runs performed with (c, d) and without (e, f) anthropogenic emissions in the study region. Red lines (e, f) depict four sub-regions used in the uncertainty analysis described in Sect. 3.4; the sub-regions contain approximately the same amounts of daily data. Note that the simulation data have been de-biased (in the same way as the data shown in Fig. 3a,b). Note also that the data which not taken into account in our inverse modeling analysis are not shown.





5    **Figure 5:** NO$_x$-to-CO$_2$ (a, b) and CO-to-CO$_2$ (c, d) emission conversion factors obtained using NO$_x$, CO, and CO$_2$ emission estimates from the EDGAR v4.2 emission inventory (a, c) and from the CDIAC and EMEP emission inventories (b, d) for the emission totals.





**Figure 6: Results of the OSSEs for estimation of NO$_x$ (a,b) and CO (c,d) emissions: the dependencies of the normalized standard error of the NO$_x$ (CO) emission estimates for the EHI (a,c) and TCO (b,d) source categories on the level of noise (that is, on the value of the diagonal elements, σ, of the error covariance matrixes) in the input "synthetic" data. The noise level is normalized to its magnitude estimated with the real measurement and simulation data. Apart from the random noise, the synthetic data included the bias that was specified (for any given day) by linearly interpolating (in time) the monthly biases shown in Fig. 3(c,d). Different colors show the results obtained with the different levels of error co-variances (ξ is the scaling factor applied to non-diagonal elements of the covariance matrix). The standard errors estimated in the "direct" way (as the RMSE representing the differences between the emission estimates inferred from the synthetic data and the "true" NO$_x$ emission estimates) and by using the subsampling technique are shown by solid and dashed lines, respectively.**





**Figure 7: The same as in and Fig. 6 but for the dependencies of the normalized standard error on the scaling factor, δ, characterizing the bias applied to the synthetic data (see Sect. 3.5 for further details).**




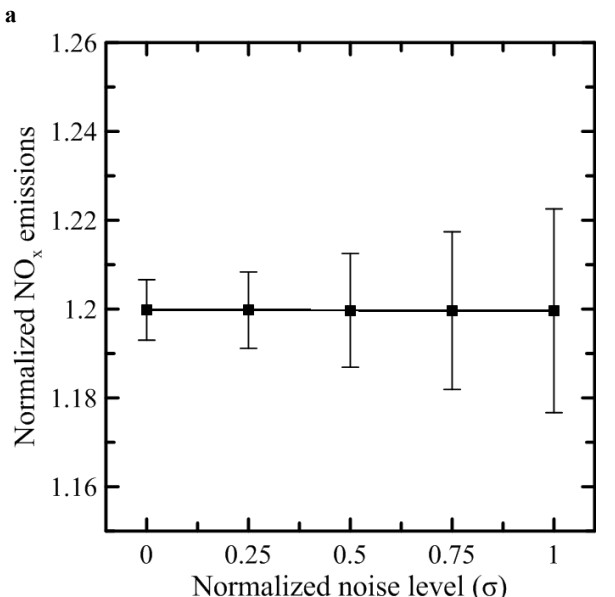
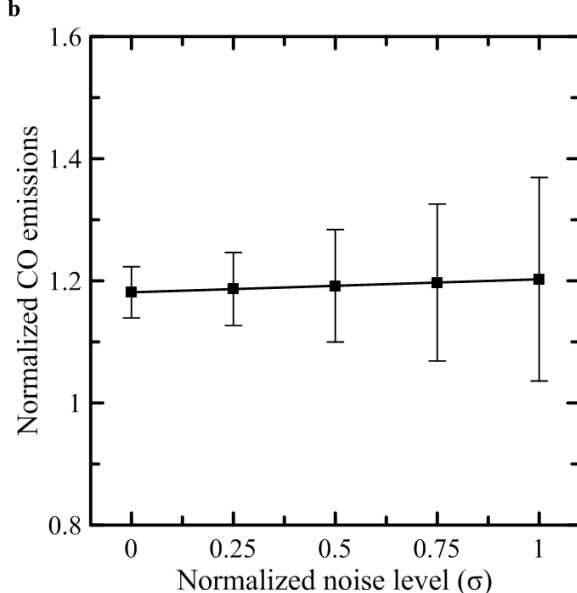

**Figure 8:** The total $NO_x$ (a) and CO (b) emission estimates ($\widehat{E}^s_{tot}$) obtained in the OSSEs where the "true" emissions were specified by scaling the bottom-up emissions (employed in the base case model run) with the factor of 1.2. The emission estimates (normalized to the respective bottom-up emission estimates, $\widetilde{E}^s_{tot}$, based on the EMEP inventory data) represent the average over the ensemble of 100 Monte Carlo experiments, each with a different sample of noise in the synthetic data, and are shown as a function the noise level ($\sigma$). Both non-diagonal elements of the error co-variance matrix and the systematic uncertainties were taken into account in the OSSEs (specifically, both $\xi$ and $\delta$ were set to be equal to unity, see further details in Sect. 3.5). Note that the value of 1.2 on the axis of ordinates corresponds to a perfect emission estimate in the case considered. Note also that the confidence intervals shown were estimated by using the subsampling technique (see Eq. 12) that is expected to predict a non-zero uncertainty (associated with the bias estimation procedure) even when the synthetic input data are not affected by random uncertainties (that is, when $\sigma$=0, see also blue lines in Fig. 7).



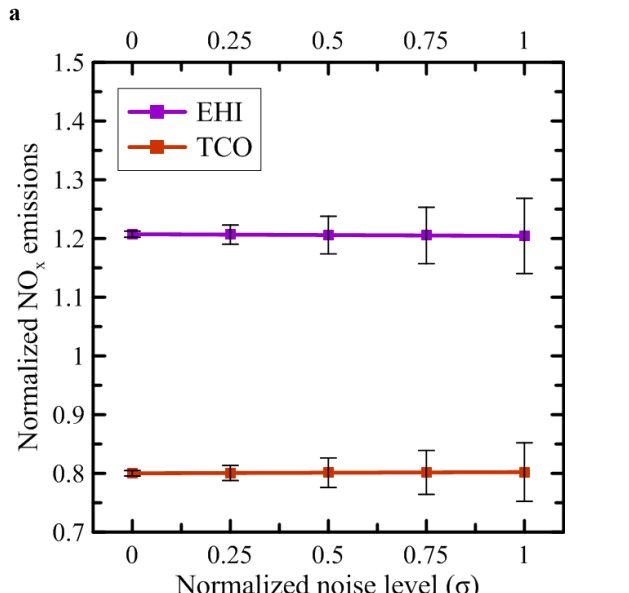

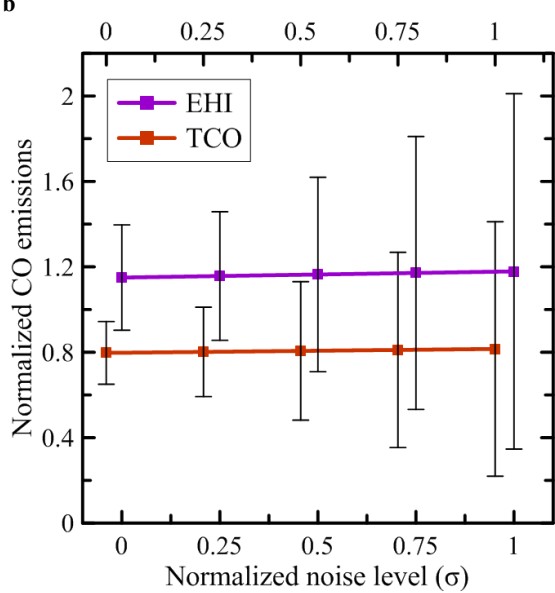

**Figure 9:** The NO$_x$ (a) and CO (b) emission estimates obtained similar to the estimates shown in Fig. 8 but separately for the two source categories (EHI and TOC) in the OSSEs where the "true" emissions in the EHI and TCO were specified by scaling the corresponding bottom-up emissions (employed in the base case model run) with the factors of 1.2 and 0.8. Note that the estimates for the EHI and TCO categories are depicted by using the abscissa axes at the bottom and at the top of the plots, respectively.



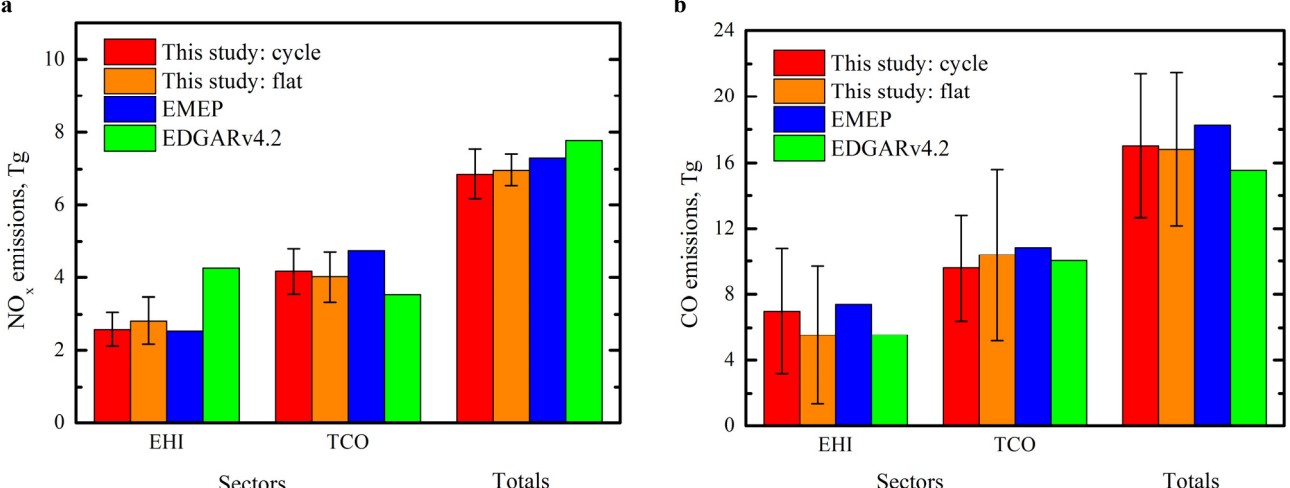

**Figure 10: Top-down estimates ($\widehat{E}_c^s$ and $\widehat{E}_{tot}^s$) of the anthropogenic NO$_x$ (a) and CO (b) emissions in the study region in comparison with the corresponding estimates from the EMEP and EDGAR v4.2 inventories. Our estimates are shown for the two cases ("cycle" and "flat") with different seasonal cycles of anthropogenic emissions.**




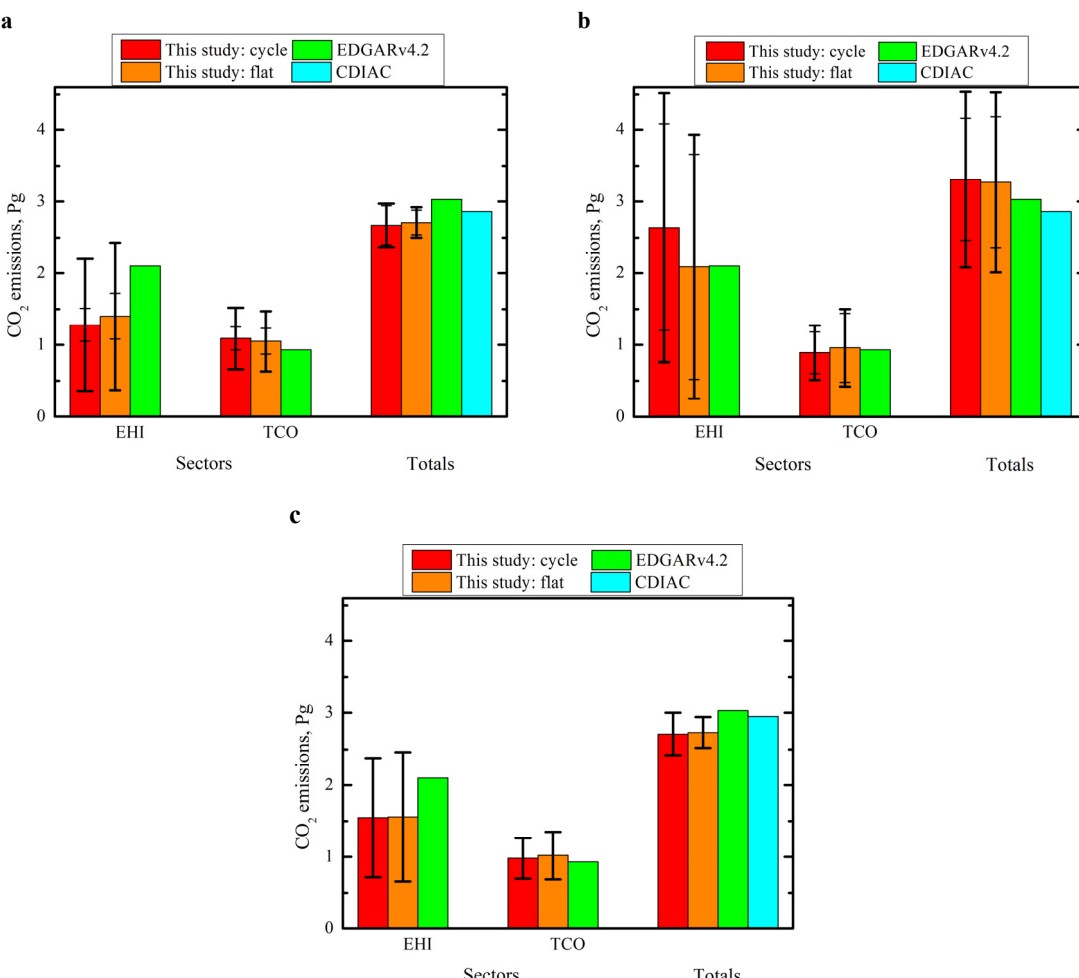

Figure 11: Hybrid estimates of the fossil-fuel $CO_2$ emissions measurements ($\widehat{E}^{co2}_{sc}$, $\widehat{E}^{co2}_{s,tot}$, $\widehat{E}^{co2}_{comb,c}$, and $\widehat{E}^{co2}_{comb,tot}$) from the study region in comparison with the data of the EDGAR v4.2 and CDIAC emission inventories. The estimates were obtained either from (a) only $NO_x$ and (b) only CO or (c) from both $NO_x$ and CO measurements. The "partial" and "full" 68.3% confidence intervals are also shown: the "partial" intervals (depicted by narrow brackets and not shown for the combined $CO_2$ emission estimates) are determined only by uncertainties in the top-down estimates of $NO_x$ or CO emissions, while the "full" intervals take into account also probable uncertainties in the conversion factors.



**Table 1.** The optimal estimates of the anthropogenic $NO_x$ and CO emissions (Tg) from the study region. The numbers in brackets represent the one-sided 68.3% confidence intervals (in percent relative to the respective optimal estimate).

| species | estim. case | EHI $\hat{E}_1^s$ | EHI EMEP | EHI EDGAR | TCO $\hat{E}_2^s$ | TCO EMEP | TCO EDGAR | Totals $\hat{E}_{sum}^s$ | Totals $\hat{E}_{tot}^s$ | Totals EMEP | Totals EDGAR |
|---|---|---|---|---|---|---|---|---|---|---|---|
| $NO_x$ | cycle | 2.59 (18) | 2.55 | 4.25 | 4.17 (15) | 4.75 | 3.53 | 6.76 (11) | 6.86 (10) | 7.30 | 7.78 |
| | flat | 2.82 (23) | | | 4.02 (17) | | | 6.84 (7) | 6.97 (7) | | |
| CO | cycle | 6.99 (54) | 7.41 | 5.55 | 9.59 (33) | 10.84 | 10.02 | 16.57 (30) | 17.03 (26) | 18.25 | 15.57 |
| | flat | 5.53 (75) | | | 10.4 (50) | | | 15.93 (25) | 16.82 (28) | | |



**Table 2.** The $NO_x$-to-$CO_2$ and CO-to-$CO_2$ emission conversion factors based on the EDGAR v4.2 emission inventory along with their relative uncertainties given in brackets as one-sided 68.3% confidence interval (in percent).

| Sectors | $NO_x$-to-$CO_2$ | CO-to-$CO_2$ |
|---------|------------------|--------------|
| EHI | 494.97 (58) | 378.63 (38) |
| TCO | 262.06 (30) | 92.42 (22) |
| TOT | 389.22 (4) | 194.50 (22) |





**Table 3.** The estimates of the fossil-fuel $CO_2$ emissions from the study region in comparison with corresponding data (when available) of the EDGAR v4.2 and CDIAC emission inventories. The numbers in brackets represent the one-sided 68.3% confidence intervals (in percent relative to the respective optimal estimate). Along with the "full" confidence intervals, the "partial" confidence intervals are shown after a slash (except for the combined estimates) that do not include uncertainties in the conversion factors.

| inversion settings | estim. case | EHI $\hat{E}_{s,1}^{CO2}$ | EHI EDGAR | TCO $\hat{E}_{s,2}^{co2}$ | TCO EDGAR | Totals $\hat{E}_{s,sum}^{co2}$ | Totals $\hat{E}_{s,tot}^{co2}$ | CDIAC | EDGAR |
|---|---|---|---|---|---|---|---|---|---|
| NO$_x$-based | cycle | 1.28 (72/18) | | 1.09 (39/15) | | 2.37 (43/12) | 2.67 (11/10) | | |
| | flat | 1.40 (74/23) | | 1.05 (40/17) | | 2.45 (44/9) | 2.71 (8/7) | | |
| CO-based | cycle | 2.64 (71/55) | 2.10 | 0.89 (42/33) | 0.93 | 3.53 (49/35) | 3.31 (37/26) | 2.86 | 3.03 |
| | flat | 2.09 (88/75) | | 0.96 (57/50) | | 3.06 (54/42) | 3.27 (38/28) | | |
| NO$_x$- and | cycle | 1.55 (54) | | 0.98 (29) | | 2.67 (33) | 2.71 (11) | | |
| CO-based | flat | 1.56 (57) | | 1.02 (33) | | 2.63 (34) | 2.73 (8) | | |