# Peer review of "Estimation of fossil-fuel CO2 emissions using satellite measurements of "proxy" species"

_Atmospheric Chemistry and Physics, 2016_

## Referee Comment (RC1) · Anonymous Referee #1 · 23 Jun 2016

General comments

This study presents a method for estimating anthropogenic CO2 emissions based on the proxy of NOx and CO emissions. The method includes rigorous quantification of uncertainties due to the uncertainty in NOx and CO emissions from inversions using satellite retrieved NO2 and CO columns, respectively and to the emission factors. The method and its explanation are generally sound, however, I have a few specific comments. I recommend the manuscript for publication with minor corrections.

Specific comments

P2, L14-15: This statement needs a bit more explanation, the 2014 emissions will be the starting point for what?

P3, L1: Technically speaking, the emissions (or emission parameters, depending on which is being optimized) are not in the transport model but are coupled to it.

Section 2.2: The authors do not mention what the boundary conditions were for other species used in the MELCHIOR scheme that react NO2 and CO. Concentrations of NO2 are strongly affected by atmospheric chemistry (lifetime of $\sim$1 day) so how important is the correct representation of chemistry on the NO2 simulations and thus in the emissions from the inversion? Similarly for CO, although owing to its longer lifetime this is perhaps not so important. How are the uncertainties in the lifetimes propagated into the emissions found from the inversions?

P9, L1-2: Why were cement emissions of CO2 ignored? What is the impact of this if the system would be used in a region where cement emissions are more important?

P9, L12-15: Do the authors mean that their motivation for solving for 2 categories of sources (EHI and TCO) is to reduced the aggregation error, because these sources have different temporal/spatial errors? If so, this could be made clearer and stated early, e.g. P9, L3 when the grouping into these categories is first mentioned.

P12, Eq. 4: How well conditioned is this expression? The authors use no regularization method?

P13, L15-16: The condition given, i.e., that the control vector is smaller than the measurement vector is not sufficient. A sufficient condition is rather the condition number of the matrix inverted. One could imagine a case where there are more measurements than unknown variables but where each measurement provides only a weak constraint (or even no constraint) on the unknown variables.

P17, L18-25: Are the results of these 2 cases discussed? They are mentioned here but there is no conclusion given about the uncertainty in the posterior due to potential errors in the seasonal cycle.

P18, L3-7: How independent are the conversion factors among the three inventories?

How are the factors determined? Do they rely on independent observations? This is important, as error in the conversion factor will translate directly into error in the CO2 emissions.

P22, L31: It would help the reader to specify again what the analysis (in section 2.4) is being referred to here, the lower sensitivity of the IASI measurements to CO emissions?

P23, L23: I would suggest either removing "robust" here or rephrasing the sentence to e.g. "...uses an approach, which is deemed more robust at the global scale", because the current formulation sounds somewhat contradictory, i.e., the EDGARv4.2 inventory is worse than EMEP in Europe but uses a more robust approach.

P25, L24-27: I'm somewhat confused. The conversion factors were determined from the ratios of CO2:NOx and CO2:CO in the inventories, so if the inversion results for NOx and CO are not significantly different from EDGARv4.2 then how can the result for CO2 be significantly different from EDGARv4.2?

Technical comments

P2, L5: "the potential"

P2, L9: "...gas, the increase in which is the driving force of recent..."

P2, L10: although I understand what the authors mean, the future tense should be used to reflect changes that will occur in the future, i.e., "...is the driving force of recent climate change and will likely remain to be in the future..."

P2, L11: "in the past decade"

P2, L17: "the compilation of"

P2, L28: "the estimation"

P3, L2: "in contrast to"

P5, L19: although I don't think all acronyms need to be specified, as some are well

known satellite names, I do think KNMI should be specified as it is an institute's name and perhaps not widely known.

P5, L31: "constraints on"

P7, L4: "allows to take into account the most important atmospheric processes"

P17, L18-19: "...emissions more probably result in..."

P22, L26: remove "quite" before "surprising"

P22, L26: "a very large number of individual retrievals"

P25, L12: I'm not sure "replicate" is really the right word here but perhaps "correspond to "

P28, L29: replace "smaller" by "lower"
* * *

---

## Referee Comment (RC2) · Anonymous Referee #2 · 6 Jul 2016

The Article *Estimation of fossil-fuel CO$_2$ emissions using satellite measurements of "proxy" species* by Konovalov et al. is an interesting study about inferring regional anthropogenic CO$_2$ emission totals from satellite measurements of NO$_2$ and CO. It is generally well written and of high scientific quality, which is why I recommend it for publication in *Atmospheric Chemistry and Physics* after minor revisions.

In particular, a revised version of the manuscript should address the following points:

- One major issue seems to be the (lack of) distinction between NO$_x$ emissions and NO$_2$ observations. It seems that the authors use NO$_x$ emission data (reported "as NO$_x$", "as N", "as NO$_2$"?) in conjunction with NO$_2$ measurements. They should explain how the uncertainty in the NO/NO2 partitioning (which can change with season and local time) influences their results, and it should be made clear to the

reader that the difference between $NO_x$ emissions and $NO_2$ observations is not problematic in this context (if this is actually the case). This is important, e.g., in the discussion of Eq. 7, and on p.21/l.14.

- The authors should comment on to what extent limiting themselves to measurements over land produces a bias in their estimates, as $NO_2$ and CO emitted over land and transported over the ocean is not considered (due to the limitation to land pixels) while the emissions are included in the emission totals. The same holds for emissions outside the study area (i.e., Ireland and Eastern Europe), which can be transported to the study area and thus be included in the measurements but not in the emission totals.

- As the present study uses the DOMINO $NO_2$ product (version 2), the reference to *Bucsela et al., 2013* given on p.5/l. 29 for the AMF uncertainties seems to be off, as that study addresses the NASA OMI NO2 product and not the DOMINO product.

- In the discussion p.10/l.32 and following, it should be stated if the same sampling (coming from cloud and intensity filtering and satellite coverage) is also applied to the CTM values. Also, it should be clearly stated that the satellite retrieval's averaging kernels are applied to all CTM profiles to get the modelled columns (at least I hope that this is the case!). Furthermore, the authors should state how they determined tropospheric columns from the CTM profiles (i.e., use of tropopause information).

- In p.11/l.8 and following, and the corresponding description in Sect. 3.2. it should be noted that the seasonal changes in the "bias" between observations and models strongly points towards systematic errors in the assumed seasonal cycle of emissions and the satellite retrievals (which mostly come from the assumed surface reflectance climatology and the emissions used in the a-priori $NO_2$ profiles used for the AMF calculations in the DOMINO retrieval).

- In the context of Eq. 3 (and in general), it would help if the authors clearly stated that their emission estimates are annual totals for the whole study region, divided by sector and species.

- Furthermore, the authors should explain how they derive the Jacobian matrix $S_c^s$.

- In Eq. 4, it should read "argmin" instead of "agrmin".

- Furthermore, in p.12/l.30, the authors should explain if and how their results are biased towards summer observations, as a result of more available satellite measurements in summer (due to cloud/intensity filtering).

- In the discussion of Eq. 6, it might help the reader if the authors would clearly state that $\Delta_i^s$ is the average difference between modelled and observed columns for the month $m$ in which observation $i$ lies. (Or did I understand this wrong? In that case, the authors should clarify their explanation of what exactly they did.)

- In Eq. 10, the authors should explicitly define $\hat{E}_{tot}^s$.

- On p.15/l.17, I believe it should read $\hat{E}_c^{CO2}omb, c$ (missing hat).

- On p.17/l.31 it might be instructive if the authors gave the sample sizes (number of daily values going into the calculations) resulting from the subsampling.

- The authors should discuss to what extent limiting themselves to only one alternative emission inventory (EDGAR and CDIAC for $NO_2$ and CO) might be problematic – after all, in principle there are more alternatives, and the authors could in principle use an ensemble of alternative inventories.

- On p.20/l.24, the authors should explain how a Cholesky decomposition (of what?) is used to create error samples.
- In p.21/l.6, it seems that the word "not" is missing in "Note that not only anthropogenic . . . ".

- The authors should explicitly state if they consider the $CO_2$ intensive cement production as part of the TCO or EHI sector.

- Comparing the "more thatn 60%" in p.28/l.24 to Fig. 10, it seems to me that this is a bit overestimated; from the figure alone it looks more like 50% to me.

- Fig. 2 should explicitly state the units on the y-axis (at least use the word "normalized" in the caption).

- Figs. 10+11 should be more specific in the units on the y-axis: $NO_x$ emissions in Tg $NO_x$ (which NO/NO2 ratio?), or Tg N, or Tg $NO_2$, . . . ? The same holds for CO and $CO_2$ emissions.

- The bar for EDGAR in Fig. 11 should be the same color as the bar for EDGAR in Fig. 10.

---

## Author Comment (AC1) · 16 Sep 2016

We are grateful to the Referee for the positive evaluation of our paper and for the useful comments which were carefully addressed in the revised manuscript. Below we describe our point-to-point responses to the referee's comments.

*Referee's comment: One major issue seems to be the (lack of) distinction between NO$_x$ emissions and NO$_2$ observations. It seems that the authors use NO$_x$ emission data (reported "as NO$_x$", "as N", "as NO$_2$"?) in conjunction with NO$_2$ measurements. They should explain how the uncertainty in the NO/NO$_2$ partitioning (which can change with season and local time) influences their results, and it should be made clear to the reader that the difference between NO$_x$ emissions and NO$_2$ observations is not problematic in this context (if this is actually the case). This is important, e.g., in the*

[Figure]

*discussion of Eq. 7, and on p.21/l.14.*

The $NO_x$ emissions are conventionally reported in inventories in grams of $NO_2$ (although actually most of them are coming in the atmosphere in the form of NO and then are being oxidized to $NO_2$). We agree that the distinction between $NO_x$ emissions and $NO_2$ observations (as well as the distinction between the species $NO_x$ and $NO_2$) was not sufficiently clear in our manuscript. Partly, it was so because we wanted to simplify our discussion, presuming (implicitly) that not only $NO_2$ columns but also $NO_x$ columns could (in principle) be retrieved from satellite measurements by using (to different extents) modeled data. In the revised version, we tried to clarify this distinction. In particular, we state (in Abstract and Introduction) that one of the proxy species considered in this study is $NO_2$ (rather than $NO_x$). Accordingly, $NO_x$ emissions are referred throughout the revised manuscript not as emissions *of* the proxy species, but rather as emissions *for* (or corresponding to) the proxy species. To clarify the role of a chemistry transport model in our study, we note (in Sect. 3.2, first paragraph) that Eq. (3) is used specifically to express the modeled relationships between $NO_2$ measurements and $NO_x$ emissions, as well as between CO measurements and CO emissions.

The uncertainties in the $NO/NO_2$ partitioning can indeed influence our results. Due to the very complex nature of this uncertainty (which may, in particular, be due to errors in the chemical reaction rates and in the boundary conditions, as well as due to inaccuracies in the reduced chemical mechanism used in our model), we cannot and did not attempt to evaluate this uncertainty explicitly. However, we expect that particularly because the $NO/NO_2$ partitioning changes with season and geographical location, its uncertainty is mostly included into the confidence intervals evaluated with the subsampling technique described in Sect. 3.4.

In the revised manuscript, the discussion of possible effects of model errors on our emission estimates is extended. In particular we note (in Sect. 3.4) that it is not quite

infeasible that some model errors associated with the representation of chemical interactions can result in similar (positive or negative) biases across the CO or $NO_x$ emission estimates inferred from the different data subsets. As an example, we mention that systematic underestimations of the $NO_x$ emissions may be due to persistent positive biases in the ozone formation rate and in boundary conditions for tropospheric ozone concentration (as ozone concentration accounts for partitioning of $NO_x$ between NO and $NO_2$) as well as due to other numerous factors (such as e.g. underestimation of the hydrocarbon emissions or of the ozone photolysis rate) that may result in underestimation of concentration of hydroxyl radical providing a major sink for $NO_x$ and determining its atmospheric lifetime. We conclude that more accurate evaluation of effects of possible errors in the model representation of chemical processes on $NO_x$ and CO emission estimates that can be derived from satellite measurements by using our inverse modeling method requires further research (involving, e.g., multi-model inversions) that goes beyond the scope of this study. Corresponding caveats are also provided in Sect 3.5 and in the final section of the manuscript (Summary and Conclusions).

Eq. (7) does not involve any emission estimates that were obtained using a model. In a corresponding discussion, we have additionally clarified the distinction between the emission estimates obtained from a bottom-up emission inventory and optimal emission estimates inferred from the measurements by using the modeled relationships between the column amounts of a given proxy species and corresponding emissions.

*Referee's comment: The authors should comment on to what extent limiting themselves to measurements over land produces a bias in their estimates, as NO2 and CO emitted over land and transported over the ocean is not considered (due to the limitation to land pixels) while the emissions are included in the emission totals. The same holds for emissions outside the study area (i.e., Ireland and Eastern Europe), which can be transported to the study area and thus be included in the measurements but*

*not in the emission totals.*

First of all, we would like to emphasize that our inversion method involves extrapolating information on pieces of the emission signature in the atmosphere, based on an atmospheric transport model, rather than simple estimation of the atmospheric budget of a proxy species. So, when removing ocean data and data outside the modeling domain, we just reduce the number of elements for such an extrapolation. Specifically, the fact that we analyzed the measurements and emissions only over land (and only over the study region) means that the measurements of $NO_2$ and CO emitted over land but transported over the ocean were not used to constrain the corresponding emissions. This limitation affected the amount of data used in the analysis (and thus the size of the vectors $\mathbf{C}_m$ and $\boldsymbol{C}_o$). However, we do not see any reason to expect that this limitation could result in any biases in our emission estimates, which would not be covered by their uncertainty intervals (evaluated as explained in Sect. 3.4). Likewise, we do not expect that any biases in our emission estimates can be caused by $NO_x$ and CO emissions outside of the study region. Indeed, on the scales considered, it seems reasonable to regard temporal and spatial variations of $NO_2$ and CO originating from any sources (including ship emission) outside of the study region as model errors on top of the modeled variations of $NO_2$ and CO originating from inside of the study region. Accordingly, we do not distinguish such variations from other errors and treat their systematic and random parts in the same ways as explained in Sect. 3.2 (see Eq. 6) and in Sect. 3.4, respectively. A corresponding discussion is provided in the revised manuscript (see Sect.3.2, the last paragraph).

*Referee's comment: As the present study uses the DOMINO NO2 product (version 2), the reference to Bucsela et al., 2013 given on p.5/l. 29 for the AMF uncertainties seems to be off, as that study addresses the NASA OMI NO2 product and not the DOMINO product.*

We agree with this remark. We presumed that the algorithms which were used to develop the NASA and KNMI data products were very similar, while, in fact, there are some noticeable differences between them. Accordingly, the reference to Bucsela et al., 2013 is replaced by the reference to Boersma et al., 2011.

*Referee's comment: In the discussion p.10/l.32 and following, it should be stated if the same sampling (coming from cloud and intensity filtering and satellite coverage) is also applied to the CTM values. Also, it should be clearly stated that the satellite retrieval's averaging kernels are applied to all CTM profiles to get the modelled columns (at least I hope that this is the case!). Furthermore, the authors should state how they determined tropospheric columns from the CTM profiles (i.e., use of tropopause information).*

Indeed, exactly the same sampling (based on the measurement information used in the satellite retrieval procedure) was applied to both the satellite data and the CTM values. Furthermore, the satellite retrieval's averaging kernels were applied to all CTM profiles that were used to get the modeled columns. In the revised version of our manuscript, the corresponding explanations (that were provided in Sect.2.2 and 2.4 of the reviewed manuscript) are extended and formulated more clearly. We also note (in Sect. 2.2) that in relatively rare cases (constituting less than 20 % of the total number of valid observations available for the study region and period) where the tropopause pressure was less than the pressure at the top of the model grid (200 hPa), the lack of the simulated data at altitudes exceeding the height of the upper model layer could result in some underestimation of the modeled tropospheric columns, but the effect of such underestimation on the results of our analysis is expected to be small, owing to application of a debiasing technique described in Sect. 3.2 and validated in Sect. 3.5.

*Referee's comment: In p.11/l.8 and following, and the corresponding description in Sect. 3.2. it should be noted that the seasonal changes in the "bias" between observa-*

*tions and models strongly points towards systematic errors in the assumed seasonal cycle of emissions and the satellite retrievals (which mostly come from the assumed surface reflectance climatology and the emissions used in the a-priori NO2 profiles used for the AMF calculations in the DOMINO retrieval).*

ACPD segment

The corresponding remark is added in Sect. 2.2 of the revised manuscript. In particular, we note (in Sect. 2.2) that the seasonal changes in the monthly biases may partly be due to errors in the seasonal cycles of the emissions specified not only in CHIMERE but also in the global models that were used to obtain the a priori $NO_2$ and CO profiles for the corresponding retrieval procedures; such changes may also be indicative of some errors in the assumed seasonal variations of other parameters of the retrieval procedures, such as, e.g., surface reflectance or atmospheric scattering by clouds and aerosol in the case of the $NO_2$ retrievals and surface temperature, local emissivity, vertical distributions of atmospheric temperature and humidity in the case of the CO retrievals.

*Referee's comment: In the context of Eq. 3 (and in general), it would help if the authors clearly stated that their emission estimates are annual totals for the whole study region, divided by sector and species.*

The components of the control vector involved in Eq. 3 are explained more clearly (as suggested by the referee) in the first paragraph of Sect. 3.2 of the revised manuscript. Note that our idea was to provide (in Sect. 3.2 and Sect. 3.3) first a description of our procedure for a general case (an arbitrary region and the arbitrary numbers of emission source categories and proxy species), since we believe that the method proposed can be used in other similar studies. Some details specific to the given study are provided in the end of Sect. 3.2 and in Sect. 3.4).

*Referee's comment: Furthermore, the authors should explain how they derive the Jacobian matrix S.*

The estimation method used in this study requires the knowledge of the product of the Jacobean matrix, S, and of the emission allocation vector, a, while the knowledge of the Jacobean matrix itself is not needed. The corresponding remark is added in the second paragraph from the end of Sect. 3.2.

*Referee's comment: In Eq. 4, it should read "argmin" instead of "agrmin".*

We are sorry for this misprint. The equation is corrected in the revised manuscript.

*Referee's comment: Furthermore, in p.12/l.30, the authors should explain if and how their results are biased towards summer observations, as a result of more available satellite measurements in summer (due to cloud/intensity filtering).*

A difference in the numbers of observations in summer and winter can result in a bias in our estimates if the assumed seasonal cycle of emissions is incorrect. For example, if the seasonal cycle overestimates the emissions in summer and underestimates in winter, then, taking into account that more satellite observations are available in summer than in winter, our annual estimates can be biased negatively. We attempted to take into account possible errors in our estimates due to errors in the temporal allocation of the emissions in the uncertainty analysis as explained in Sect. 3.4. A corresponding remark is added in Sect. 3.2 (see the paragraph before Eq. 6).

*Referee's comment: In the discussion of Eq. 6, it might help the reader if the authors would clearly state that $\Delta_i^s$ is the average difference between modelled and observed columns for the month m in which observation i lies. (Or did I understand this wrong? In that case, the authors should clarify their explanation of what exactly they did.)*

[Figure]

Indeed, $\Delta_i{}^s$ is the average difference between the modelled and observed columns for the month $m$ in which observation $i$ lies. An explanation provided before Eq. (6) is revised accordingly.

*Referee's comment: In Eq. 10, the authors should explicitly define .*

The explanation that was provided after Eq. (10) in the reviewed manuscript is revised and is defined explicitly.

*Referee's comment: On p.15/l.17, I believe it should read (missing hat).*

We are sorry for this misprint which is corrected in the revised manuscript.

*Referee's comment: On p.17/l.31 it might be instructive if the authors gave the sample sizes (number of daily values going into the calculations) resulting from the subsampling.*

The requested numbers are provided in Sect. 3.4 of the revised manuscript.

*Referee's comment: The authors should discuss to what extent limiting themselves to only one alternative emission inventory (EDGAR and CDIAC for NO2 and CO) might be problematic after all, in principle there are more alternatives, and the authors could in principle use an ensemble of alternative inventories.*

Ideally, it would indeed be best to consider an ensemble of several independent inventories providing the data on spatial distributions of emissions of all the species ($NO_x$, CO and $CO_2$) involved in our analysis. We tried but, apart from the data of the EDGAR inventory, we could not find publicly open inventory data satisfying these criteria and available for the region and period considered. In particular, we examined several in-

ventories presented in the GEIA data base (http://eccad.sedoo.fr/). So, in this study, in view of the limited practical availability of necessary inventory data, the "ensemble" approach could not be fully realized. A corresponding remark is added in Sect. 3.4 of the revised manuscript. Limitations of a simpler and slightly different approach used in this study are in detail discussed in Sect. 3.4.

*Referee's comment: On p.20/l.24, the authors should explain how a Cholesky decomposition (of what?) is used to create error samples.*

The requested explanation is added in Sect. 3.5 of the revised manuscript. In particular, we explain that samples of the errors with the covariance structure given by Eq. (16) were generated from a Gaussian distribution by using a standard method (Press et al., 1992) involving the Cholesky decomposition of the correlation matrices that were specified, in our case, using the covariance functions given by Eqs. (16) and (17). The Cholesky decomposition of a correlation matrix gives a lower-triangular matrix, **L**; applying this matrix to a vector of uncorrelated samples of Gaussian noise, $u$, gives a vector, **L**$u$, with the components satisfying the original correlation matrix.

*Referee's comment: In p.21/l.6, it seems that the word "not" is missing in "Note that not only anthropogenic. . . ".*

Indeed, the word "not" was missing. A corresponding correction is made in the revised manuscript.

*Referee's comment: The authors should explicitly state if they consider the $CO_2$ intensive cement production as part of the TCO or EHI sector.*

We stated in Sect. 2.3 that $CO_2$ emissions from cement production are not considered

in our study. A reason is that, unlike FF burning, cement production is not associated with considerable emissions of either $NO_x$ or CO, and so satellite measurements of the corresponding proxy species cannot provide strong constraints on $CO_2$ emissions from cement production.

*Referee's comment: Comparing the "more than 60%" in p.28/l.24 to Fig. 10, it seems to me that this is a bit overestimated; from the figure alone it looks more like 50% to me.*

Actually, we say about 60% relative to our estimates, not relative to the EDGAR data. A clarifying remark is added in the revised manuscript.

*Referee's comment: Fig. 2 should explicitly state the units on the y-axis (at least use the word "normalized" in the caption).*

It is indicated in the revised manuscript, that the values shown in Fig 2 are the normalized monthly $NO_x$ and CO emissions and are unitless.

*Referee's comment: Figs. 10+11 should be more specific in the units on the y-axis: $NO_x$ emissions in Tg $NO_x$ (which $NO/NO_2$ ratio?), or Tg N, or Tg $NO_2$, . . . ? The same holds for CO and $CO_2$ emissions.*

We indicated that $NO_x$ emissions on the y-axis of Figs 10 and 11 are given in Tg $NO_2$ and $CO_2$ emission are given in Pg $CO_2$.

*Referee's comment: The bar for EDGAR in Fig. 11 should be the same color as the bar for EDGAR in Fig. 10.*

Actually, the both figures were plotted using exactly the same color settings. To avoid a

possible impression (which may be due to differences in surroundings) that green color used in the bar for EDGAR in Fig. 10 is darker than that in Fig. 11, the figures are re-plotted using a different color scheme.

---

## Author Comment (AC2) · 16 Sep 2016

We thank the Referee for the positive evaluation of our manuscript and for the thoughtful comments and remarks. All of the referee's comments have been carefully addressed in the revised manuscript. Below we describe our point-to-point responses to the referee's comments.

*Referee's comment: P2, L14-15: This statement needs a bit more explanation, the 2014 emissions will be the starting point for what?*

We meant that the 2014 emissions would be the starting point for global and national emission reduction plans. However, this sentence is not quite necessary in the given context and is removed from the revised version of our manuscript.

*Referee's comment: P3, L1: Technically speaking, the emissions (or emission parameters, depending on which is being optimized) are not in the transport model but are coupled to it.*

We agree with this remark. The corresponding sentence is corrected accordingly.

*Referee's comment:* Section 2.2: *The authors do not mention what the boundary conditions were for other species used in the MELCHIOR scheme that react $NO_2$ and CO. Concentrations of $NO_2$ are strongly affected by atmospheric chemistry (lifetime of $\sim 1$ day) so how important is the correct representation of chemistry on the NO2 simulations and thus in the emissions from the inversion? Similarly for CO, although owing to its longer lifetime this is perhaps not so important. How are the uncertainties in the lifetimes propagated into the emissions found from the inversions?*

It was mentioned in the reviewed manuscript (Sect. 2.2) that initial and boundary conditions for gases and aerosols were specified using monthly climatological data from LMDz-INCA global model. In the revised manuscript (Sect. 2.2), we addressed this point in more detail. In particular, we indicated several concrete species, for which the boundary conditions were specified using the LMDz-INCA data and noted that a full list of such species is provided in the CHIMERE documentation available on the web site www.lmd.polytechnique.fr/chimere. We also noted that influxes of other species, most of which are very reactive and short-lived (such as, e.g., OH and $HO_2$) into a model domain, are not specified in CHIMERE.

It is true that the correct representation of chemistry in the $NO_2$ simulations is an important prerequisite for inferring reliable $NO_x$ emission estimates. The effect of uncertainties in simulations of chemical processes on our results was briefly discussed in Sect. 3.4 (P. 17, L. 13-17) of the reviewed manuscript. In particular, we argued that

such model errors are likely to differ in different grid cells, and thus they are expected to contribute to the emission estimate uncertainties evaluated using the subsampling technique. In the revised manuscript, the discussion of this important point is improved and extended. In particular, we added that as the CO and $NO_2$ evolution is governed by essentially different chemical processes, uncertainties due to a "chemical" part of model errors are likely to be manifested in differences between the $CO_2$ emission estimates based on the $NO_2$ and CO measurements. But we also noted that it is nonetheless not quite infeasible that some model errors associated with the representation of chemical interactions can result in similar (positive or negative) biases across the CO or $NO_x$ emission estimates inferred from the different data subsets, and so we cannot completely ensure that the confidence intervals for our CO and (especially) $NO_x$ emission estimates actually account for all possible model errors. A similar caveat is provided in Sect. 5 (Summary and conclusions) of the revised manuscript.

In fact, the effects of "chemical" part of model errors on inversion results are very difficult to evaluate as the characteristics of such errors are mostly not known and probable errors in specific reactions or in the boundary conditions of certain species may constitute practically an infinite number of error combinations which would propagate into the emissions found from the inversions in different ways. So we believe that more accurate evaluation of the effects of possible errors in the model representation of chemical processes on $NO_x$ and CO emission estimates that can be derived from satellite measurements requires further research going beyond the scope of this study. One of the practical ways to address this issue could involve multi-model inversions that might be performed in the framework of a dedicated project. The results of our study can provide an impetus for such a project.

*Referee's comment: P9, L1-2: Why were cement emissions of $CO_2$ ignored? What is the impact of this if the system would be used in a region where cement emissions are more important?*

Emissions from cement production are not considered in our study mostly because cement production, unlike FF burning, is not associated with significant emissions of either $NO_x$ or CO, and so satellite measurements of the corresponding proxy species cannot provide strong constraints on cement emission of $CO_2$. A corresponding explanation is added in the revised manuscript. Even if cement $CO_2$ emissions would be more important in the case considered, our estimates of FF emissions would not be directly affected (again because cement production do not contribute to concentrations of the proxy species and corresponding emissions were excluded from our calculations of the conversion factors).

*Referee's comment: P9, L1-2: P9, L12-15: Do the authors mean that their motivation for solving for 2 categories of sources (EHI and TCO) is to reduced the aggregation error, because these sources have different temporal/spatial errors? If so, this could be made clearer and stated early, e.g. P9, L3 when the grouping into these categories is first mentioned.*

Yes, our primary motivation for defining the two specific categories was to limit aggregation errors, but we expected also that consideration of these two categories would allow us to get more specific information on emission sources. The corresponding paragraph was revised accordingly.

*Referee's comment: P12, Eq. 4: How well conditioned is this expression? The authors use no regularization method?*

Indeed, we did not use any regularization method (or a priori constraints on the solution). The main reason is that the number of control parameters is very small (one or two) especially if considering the vast amount of atmospheric data which we use. This reason was briefly explained on P. 13 of the revised manuscript, and we tried to improve the corresponding discussion in the revised manuscript. Another reason is that we presume that each control variable is seen with some level of independence

by subsets of these data given the geographical distribution of each. The results of our OSSEs based on the Monte Carlo method (see Sect. 3.5) show that the uncertainties in our emission estimates remain rather small in spite of very large uncertainties in the input data (see Sect. 2.4). In our understanding, this fact clearly indicates that the inverse problem considered is not ill-conditioned. This is mentioned in Sect. 3.5. The analysis of the relationship between the CO or $NO_x$ measurements and the emission estimates by using methods of linear algebra was beyond the scope of this complex and time-consuming study (and would be pointless in the important case where the control vector has only one component corresponding to the total $NO_x$ or CO emissions). Application of such methods to the inverse problem considered in our study is complicated due to the fact that the cost function given by Eq. (4) includes the "bias" term that depends on the model data (and thus on the emission estimates) in a rather complex manner (according to Eq. 6).

*Referee's comment: P13, L15-16: The condition given, i.e., that the control vector is smaller than the measurement vector is not sufficient. A sufficient condition is rather the condition number of the matrix inverted. One could imagine a case where there are more measurements than unknown variables but where each measurement provides only a weak constraint (or even no constraint) on the unknown variables.*

We fully agree with this critical remark. The corresponding text is corrected accordingly. Specifically, instead of the incorrect statement given in the reviewed manuscript, we mention that avoiding formal a priori constraints (or any other regularization) does not *necessary* result in ill- conditioning of an inverse problem, as long as the dimension of the control vector is much smaller than that of the measurement vector. We mention also that although satisfying this criterion alone cannot guarantee that the problem is well-conditioned, the numerical experiments presented below in Sect. 3.5 show that errors in our emission estimates due to probable errors in input data remain limited and thus the results of these experiments indicate that the problem considered in this study

is not ill-conditioned.

*Referee's comment: P17, L18-25: Are the results of these 2 cases discussed? They are mentioned here but there is no conclusion given about the uncertainty in the posterior due to potential errors in the seasonal cycle.*

Yes, the results of these two cases are discussed in Sect. 4.1 (in the $5^{th}$ paragraph from the beginning of the section) and in Sect. 4.2 (in the last paragraph).

*Referee's comment: P18, L3-7: How independent are the conversion factors among the three inventories? How are the factors determined? Do they rely on independent observations? This is important, as error in the conversion factor will translate directly into error in the CO2 emissions.*

The $NO_x$-to-$CO_2$ (or CO-to-$CO_2$) emission conversion factors were calculated in our study as the ratios of the corresponding emission annual totals provided by the emission inventories according to Eq. (7). In Sect. 3.4, we explain that we used two different sets of estimates for the conversion factors, one of which (considered as the main option) is based entirely on the EDGARv4.2 emission inventory, while another is based on the data for $CO_2$ emissions from CDIAC and for $NO_x$ and CO emissions from the EMEP inventory. Therefore, we presume that the first question of the reviewer (*How independent are the conversion factors among the three inventories?*) can be reformulated as follows: are the differences between the $CO_2$ emission estimates given by the CDIAC and EDGARv4.2 inventories and the differences between the $NO_x$ (CO) emission estimates provided by the EDGARv4.2 and EMEP inventories sufficiently representative of uncertainties in the inventory data (as assumed in our study)? Although we cannot provide a mathematically precise answer to this question, the fact is that there are considerable differences in both the data sources and the methodologies used across the three inventories. Specifically, while the fossil fuel burning $CO_2$ emission data provided by CDIAC are based on the energy statistics published by the

United Nations (UN, 2012), the EDGARv4.2 inventory uses energy activity data based on IEA (International Energy Agency) energy balances (IEA, 2012). The UN data used in CDIAC were compiled primarily from the annual energy questionnaire distributed by the United Nations Statistics Division and supplemented by official national statistical publications (UN, 2012), while the IEA data were compiled following harmonised definitions and comparable methodologies across countries and do not necessary represent complete data sets available to national experts (IEA, 2010). Similarly, the methodologies used in CDIAC and EDGARv4.2 to convert fuel consumption into $CO_2$ emissions have been developed independently and involve significantly different classifications of fuels and different sets of parameters. For example, while the key parameter involved in the EDGARv4.2 inventory is the net caloric value which is used to convert the activity data into the energy values (IEA, 2010) (that are then converted into quantities of carbon), CDIAC converts the quantity of fuel into the quantity of carbon directly by using the carbon content parameter (see Marland and Rotty, 1984 and IPCC, 2006 for more details on the methodologies used in the CDIAC and EDGARv4.2 inventories, respectively). The EMEP/CEIP inventory is based on emission reports provided by the national environmental agencies. Accordingly, compared to the EDGARv.4.2 inventory, the emission data provided by EMEP inventory may better account for statistical information and sources specific for a given country. The set of emission factors which EMEP recommends to use while preparing national emission inventories (EMEP/EEA, 2013) is substantially different from that used in the EDGAR v4.2 inventory (IPCC, 2006), particularly because it involves the different sector definitions. Taking all these differences into account, we believe that it is indeed safe to assume that the two kinds of the conversion factor estimates considered in our study are sufficiently independent. Nonetheless, it is also not quite impossible that, in some hypothetical cases (but hardly in our study region), different inventories can be biased in a similar way due to, e.g., the sources and technologies that are accounted for neither in international nor in national energy data bases. In such cases, the conversion factor uncertainty evaluated with our approach may be underestimated. So, in a general situation, a statistically significant difference between our "hybrid" $CO_2$ emission estimate and an estimate based on emission inventory data would strongly suggest that the latter is biased, although other, less probable, reasons, such as errors is the conversion factors or systematic uncertainties in the model representation of chemical processes should not be disregarded without special investigation.

Consistently with the above discussion, we mentioned in the revised manuscript (without going into details) that taking into account considerable differences in the data sources and methodologies used across the three inventories, we assume that the main and alternative conversion factor estimates are sufficiently independent. A similar assumption concerning reliability of the confidence intervals is also mentioned in Sect. 4.2. Finally, we remarked in Sect. 5, that further research is needed to ensure that the confidence intervals for our emission estimates actually take into account all possible error sources, including uncertainties in the conversion factors.

*Referee's comment: P22, L31: It would help the reader to specify again what the analysis (in section 2.4) is being referred to here, the lower sensitivity of the IASI measurements to CO emissions?*

We added an explanation. The corresponding revised sentence reads as follows: "Taking into account our preliminary analysis (see Sect. 2.4) indicating that the contribution of the anthropogenic CO emissions in the study region into the corresponding CO columns is relatively small and the results of the OSSEs (see Sect. 3.5), it is also not surprising that the uncertainties in our CO emission estimates are much larger than those in the $NO_x$ emission estimates."

*Referee's comment: P23, L23: I would suggest either removing "robust" here or rephrasing the sentence to e.g. ". . .uses an approach, which is deemed more robust at the global scale", because the current formulation sounds somewhat contradictory,*

*i.e., the EDGARv4.2 inventory is worse than EMEP in Europe but uses a more robust approach.*

The corresponding sentence has been rephrased following the referee's suggestion and reads as follows: "This is an expected result because the methodology used in the EMEP inventory is specific to national statistical data available from European countries, while the EDGAR v4.2 inventory uses another approach which is deemed to be robust at the global scale."

*Referee's comment: P25, L24-27: I'm somewhat confused. The conversion factors were determined from the ratios of $CO_2$:$NO_x$ and $CO_2$:CO in the inventories, so if the inversion results for $NO_x$ and CO are not significantly different from EDGARv4.2 then how can the result for $CO_2$ be significantly different from EDGARv4.2?*

Actually, the inversion results for $NO_x$ are significantly different from EDGARv4.2, but this was not explicitly stated in the reviewed manuscript. This point is clarified in Sect. 4.1 of the revised manuscript.

The technical comments by referee were carefully considered. The corresponding changes are made in the revised manuscript.

**References**

UN: 2012 Energy Statistics Yearbook. United Nations Department for Economic and Social Information and Policy Analysis, Statistics Division, New York, 2012.

IEA: Energy Statistics of OECD and Non-OECD Countries. On-line data service. URL: data.iea.org , 2012.

IEA: $CO_2$ Emissions from Fuel Combustion 2010, OECD Publishing, Paris.DOI: http://dx.doi.org/10.1787/9789264096134-en, 2010.

Marland, G. and Rotty, R. M.: Carbon dioxide emissions from fossil fuels: a procedure for estimation and results for 1950–1982, Tellus B, 36B, 232–261, doi:10.1111/j.1600-0889.1984.tb00245.x, 1984.

IPCC: 2006 IPCC Guidelines for National Greenhouse Gas Inventories, IPCC National Greenhouse Gas Inventory Programme, available at: http://www.ipcc-nggip.iges.or.jp/public/2006gl/index.html, Hayama, Japan, 2006.

EMEP/EEA: EMEP/EEA Air Pollutant Emission Inventory Guidebook 2013. Technical report No. 12/2013, August 2013. Copenhagen: European Environment Agency, 2013.